# Measurement report: Microphysical and optical characteristics of radiation fog - a study using in-situ, remote sensing, and balloon techniques.

Katarzyna Nurowska<sup>1,2</sup>, Przemysław Makuch<sup>3</sup>, and Krzysztof M. Markowicz<sup>1</sup>

20

**Correspondence:** Katarzyna Nurowska (Katarzyna.Nurowska@fuw.edu.pl)

**Abstract.** Based on in situ observations, remote sensing, and tethered balloon soundings, this study investigates the vertical profiles of microphysical and thermodynamic properties within radiative fog layers in the Strzyżów valley (Southeastern Poland). Across three case studies of radiation fog in September 2023, 74 soundings were performed, with 41 employing the OPC-N3 instrument to capture droplet spectra. The results indicated similar weather conditions in all cases, with a liquid water path consistently above 15 g·m<sup>-2</sup>, most observations remained within the thin fog regime. The effective droplet radius decreased with height (between 3-4.6  $\mu$ m for 100 m), with larger droplets (18.5  $\mu$ m) concentrated near the ground.

Fog dissipation occurred simultaneously from both the top and bottom. The mature fog stage was characterized by peaks in both liquid water content (LWC) and droplet number concentration ( $N_c$ ), typically located at approximately 80% of the total fog depth. Theoretical estimates of terminal velocity for droplets  $\geq 18.5~\mu m$  suggest that larger droplets are removed from fog layers within minutes, affecting the longevity of the fog. Calculated values of equivalent adiabaticity values ( $\alpha_{eq}$ , defined as the scaling factor by which the adiabatic lapse rate of the mixing ratio needs to be multiplied to match the observed liquid water path) ranged from 0 to 0.6. Except for a single case where near-ground  $\alpha_{eq}$  was negative — a phenomenon that is seldom documented in the fog literature.

Having instruments measuring radiation at two different heights (below and above the fog layer), it was possible to estimate the effect of fog on reducing total shortwave and longwave (NET - downward - upward) radiation at ground level. Comparisons of NET radiation across the fog layer before and after its dissipation revealed differences of up to 150 W·m<sup>-2</sup>. A linear relationship was observed between the reduction in longwave radiation and the liquid water path in our measurements; however, since all examined fog events were predominantly optically thin, this finding should be interpreted as applying primarily to optically thin fog conditions.

As a result of the measurements, average values of liquid water content and droplet number concentrations were obtained for the observed optically thin fogs in the valley area. Mean LWC in the fog layer core was found between 0.2– $0.4 \, \rm g \cdot m^{-3}$ , with  $N_c$  up to  $300 \, \rm cm^{-3}$ . The near-surface mean effective radius ranged between 8 and  $10 \, \mu m$  and exhibited a linear decrease with height. The good agreement between radiative transfer model outputs and observed radiative fluxes supports the accuracy of the retrieved microphysical parameters used as model inputs.

<sup>&</sup>lt;sup>1</sup>University of Warsaw, Faculty of Physics, Institute of Geophysics

<sup>&</sup>lt;sup>2</sup>Guangdong Technion - Israel Institute of Technology, Department of Mathematics

<sup>&</sup>lt;sup>3</sup>Polish Academy of Sciences, Institute of Oceanology

## 25 1 Introduction

55

A characteristic feature of radiation fogs is their localized nature, as they do not cover large areas, making their forecasting challenging. Weather conditions contribute to approximately 30% of aviation accidents in the USA (Gultepe, 2023). Radiation fog significantly reduces visibility and complicates navigation, posing a threat to transportation. According to the American National Transportation Safety Board (NTSB), fog is the second most critical weather-related factor leading to fatal aviation accidents, accounting for an estimated 14% of such incidents (Capobianco and Lee, 2001). Fog affects not only safety but also imposes significant economic costs. It can disrupt road traffic, force ships to alter their routes, and result in airport closures. In the United States, weather is the leading cause of aircraft delays, accounting for over 70% of all cases (Kasper, 2016). Among weather-related factors, low visibility and low cloud ceilings are major contributors, as they require increased spacing between landing aircraft to maintain safety, thereby reducing airport throughput. According to NTSB data, visibility-related conditions contribute to approximately 30–35% of all flight cancellations (Stevens, 2019; Gultepe, 2023).

Fog is a meteorological phenomenon occurring near the Earth's surface, characterized by the suspension of water droplets in the air, significantly reducing visibility to below 1 km (George, 1951). Several types of fog exist, depending on their formation mechanisms. This article focuses on radiation fog, which primarily forms at night under clear-sky and minimal wind conditions, within a stable boundary layer (SBL). Under such conditions, the ground surface cools significantly, leading to the cooling of the air immediately above it (Lakra and Avishek, 2022). Once the dew point temperature is reached, water vapor condenses on suspended particles (condensation nuclei), forming fog. This type of fog develops from the ground upwards, usually not exceeding 200 meters in height. The cooling of successive air layers occurs from the lower layer upward, which is why radiation fogs are associated with the formation of temperature inversions. After sunrise, and with the onset of stronger winds, the fog and the inversion dissipate. When radiation fog forms, it initially remains optically thin to longwave (LW) radiation and develops within a stable lapse rate. When fog becomes optically thick, cooling occurs predominantly at the top of the fog layer, while the portion near the ground radiates in the LW range that is able to warm the surface (Mason, 1982; Price, 2011). The potential equivalent temperature becomes uniform throughout the fog layer, inducing slight instability, which in turn increases turbulence within the fog. As demonstrated by Price (2011), approximately 50% of the fog cases he analyzed transitioned into optically thick, well-mixed fogs characterized by a saturated adiabatic stability profile. His research suggests that this conversion typically occurs when the fog layer exceeds 100 meters in thickness. Numerical weather models have difficulty catching the shift from optically thin to optically thick fog (Poku et al., 2021; Boutle et al., 2022; Antoine et al., 2023).

Costabloz et al. (2024) studied fog development during the SOFOG3D experiment. They proposed several methods to identify the point at which the transition from thin to optically thick fog occurs:

- surface LW net radiation should approach 0. In their research they assumed that this condition occurs when  $|LW_N| < 5 \text{ W} \cdot \text{m}^{-2}$ ,
- the air temperature profile within the fog layer should decrease with height, as the air near the surface is warmed by the ground while the fog top cools radiatively. They checked whether the temperature gradient was negative by comparing temperatures at 25 m and 50 m,

- turbulent kinetic energy exceeds  $0.10 \text{ m}^2 \cdot \text{s}^{-2}$ ,
- fog top height exceeds 110 m,
  - Wærsted et al. (2017) proposed that a transition to optically thick fog occurs when LWP>30 g·m<sup>-2</sup>, but Costabloz et al. (2024) suggested a value of  $15 \text{ g·m}^{-2}$  to match more closely the time when the other criteria are met.

Those conditions were met in the SOFOG3D experiment within about 1 hour.

Key factors influencing the likelihood of fog transitioning into an optically thick state include the time of its formation (the earlier before sunrise, the more likely) and the humidity profile of the air (Boutle et al., 2018). For droplets to begin forming, aerosols acting as cloud condensation nuclei (CCN), such as ammonium nitrate aerosols, are required. In clouds, turbulence can uplift air masses, activating CCNs more rapidly and extensively. In fog, droplet growth is primarily governed by radiative cooling. As demonstrated by Boutle et al. (2018), a higher concentration of large aerosol particles accelerates the transition to a well-mixed fog state. Additionally, the type of aerosol present in the air is important; compounds with high hygroscopicity that can activate at low supersaturation levels are most effective as CCN (Gilardoni et al., 2014).

According to Costabloz et al. (2024), during the SOFOG3D, inverted LWC profiles—maximum LWC found at the ground and decreasing with altitude—were commonplace in optically thin fogs. Mostly in well-mixed optically thick fogs, quasi-adiabatic profiles with LWC increasing with height were found. However, in one case, they measured LWC profiles decreasing with height one hour after the transition occurred and LWC values at the ground reached 0.25 g.m-3, the highest values recorded during the whole campaign.

Research utilizing cloud radars, ceilometers, and microwave radiometers has made it possible to establish the rate at which LW radiative cooling at the top of the fog layer can lead to condensation within the fog. Under clear-sky conditions, when the liquid water path (LWP) exceeds  $30~{\rm g\cdot m^{-2}}$ , this cooling (above the fog) can result in the formation of liquid water at a rate of up to  $70~{\rm g\cdot m^{-2}\cdot h^{-1}}$  (Wærsted et al., 2017).

The presence of clouds above the fog can also influence water condensation, with low clouds potentially blocking cooling entirely, leading to fog dissipation.

After sunrise, shortwave (SW) radiation begins to heat the fog, causing droplet evaporation. Wærsted et al. (2017) estimated that the strength of this process is about 10-15 g·m<sup>-2</sup>·h<sup>-1</sup>. The rate of evaporation increases with the effective radius of droplets ( $r_{eff}$ ) and LWP, and decreases with larger solar zenith angles. Additionally, the warming of the ground surface transfers approximately 30 g·m<sup>-2</sup>·h<sup>-1</sup> of sensible heat to the fog.

To accurately predict the formation and evolution of fog, a weather forecasting model must effectively represent the interactions between the atmosphere and the Earth's surface, various processes (such as microphysics, radiation, and turbulence), and it must do so on a local scale while accounting for terrain features.

One approach to studying fog is through large-eddy simulations (LES) modeling. This approach enables the examination of turbulence effects and interactions between the atmosphere and the surface (Maronga and Bosveld, 2017), the deposition of droplets on vegetation (Mazoyer et al., 2017), or the influence of the urban canopy (Bergot et al., 2015) on fog formation and evolution. Numerical models often struggle to accurately forecast fog formation, dissipation, depth, or water content

(Román-Cascón et al., 2012; Zhou et al., 2012; Bari et al., 2023). This difficulty arises from the fog's localized nature and the delicate balance between processes such as radiation balance, droplet deposition on the surface, turbulent mixing, microphysical properties, and moisture availability. Recently, AI-based tools, including machine learning and deep learning, have been employed to enhance numerical weather prediction (NWP). While these methods have shown promising results, they also introduce new challenges. Machine learning requires high-quality datasets specific to each forecast location, as well as substantial computational resources to produce timely results (Bari et al., 2023).

For the initialization of numerical models or the development of methods to retrieve LWP from satellites, it is essential to understand the microphysical properties of fog as a function of height. Unfortunately, there is a scarcity of data on the vertical distribution of fog's microphysical characteristics. Measurements using aircraft are impractical because fog typically forms close to the Earth's surface and inherently reduces visibility. However, measurements can be conducted using aerological balloons Egli et al. (2015), instrumentation placed on tall towers Ye et al. (2015); Han et al. (2018), and more recently, drones and microwave radiometers (MWR) have become viable options for such observations.

100

105

120

Using a tethered balloon, Pinnick et al. (1978) made the first measurements of the vertical profiles of microphysical characteristics in fog. They showed that in the studied cases, a fog had a bimodal distribution of droplets (r=5  $\mu$ m and r=0.6  $\mu$ m) with LWC range from  $10^{-4}$  to 0.45 g·m<sup>-3</sup>.

Egli et al. (2015) performed soundings with a tethered balloon, and measured LWC,  $N_c$  and  $r_{eff}$  every 10 m. His results from two fog cases show that the changes in LWC are related to the change in  $N_c$  and not to the change in droplet size. In most cases,  $r_{eff}$  was constant with height. One fog case was characterised by low LWC (maximum of 0.14 g·m<sup>-3</sup>), however, high  $N_c$  above 2000 cm<sup>-3</sup>. In this case of fog, 3 measurements were taken. Omitting the values of  $r_{eff}$  at the very bottom of the profiles (where the values dropped significantly), the value of  $r_{eff}$  decreased with height. In the case of one profile, the value of eff at a 25 m reached a maximum of 9.4  $\mu$ m. The second fog case, with six soundings, consisted of a considerably thicker fog with higher LWC and  $r_{eff}$  values, although accompanied by lower total drop counts. The LWC had a constant pattern in the first third of the height, then LWC increased with height, and then decreased with height to the cloud top. The highest LWC value was 0.54 g·m<sup>-3</sup>.  $N_c$  had a similar pattern with height as LWC. The highest  $N_c$  value recorded was 500 cm<sup>-3</sup>. The  $r_{eff}$  values differ from sounding to sounding, however they were constant with height, in range between 4 and 8  $\mu$ m.

The motivation for this study is the miniaturization of equipment for particle detection. For example, the Alphasense OPC-N3 - optical particle matter (PM) sensor, commonly used for aerosol monitoring, can also be used to measure the microphysical properties of fog when mounted on a tethered balloon or drone (Nurowska et al., 2023). Such a system was employed to capture vertical profiles of radiative fog in a mountain valley, a region where air pollution can be elevated during inversion conditions. This type of terrain enables fog monitoring at different altitudes. In this setup, SW and LW radiometers positioned near the valley bottom and mountain top allow determination of the optical, microphysical, and radiation closure of the fog. Section 2 outlines the instruments utilized for conducting the measurements, while Section 3 details the methodology of the in-situ measurements and the model setup. The core of the article is presented in Section 4, which features a case study of radiative fog occurrence, including optical, microphysical, and radiation closure analyses performed for this case. Section 4 focuses on an event in the Strzyżów valley, where data were gathered using a balloon. The 1D Fu-Liou radiative transfer model (Fu

and Liou, 1992, 1993) was applied to simulate the conditions in the Strzyżów valley, incorporating additional data from the SolarAOT station (which consists of an upper and lower station).

## 130 2 Experiment setup

140

This study is based on measurements taken at two sites in Strzyżów. This small town is located in southern Poland, in the region of the Strzyżowskie foothills. The city is located next to the Wisłok River. The research was conducted using remote sensing and in situ techniques as well as by an apparatus connected to a tethered balloon. In addition, numerical simulations were used for the radiation closure study.

# 135 2.1 SolarAOT<sub>lower</sub> - launching site

The lower station is located in the valley of Strzyżów city at 260 m.a.s.l. (49°52'18.0"N 21°48'26.0"E). On the site of balloon launching, there was mounted a CNR4 net radiometer for upward and downward SW and LW flux; a meteo station including MetPak and sensors A100LK, W200P, HYT936, and OPC-N3. In addition, the mobile laboratory equipped with Aurora 4000 nephelometer, Laser Aerosol Spectrometer LAS 3340A, and Oxford Lasers VisiSize D30 (ShadowGraph) was used at this site. Raymetrics single-wavelength (532 nm) lidar 510M for aerosol and cloud detection was used.

The VisiSize D30 system, developed by Oxford Lasers Ltd., operates using the shadowgraph technique. The VisiSize D30, hereafter referred to as ShadowGraph, captures shadow images of particles as they pass through the measurement volume between a laser head and a high-resolution camera. This system enables the determination of microphysical properties, including particle shape, size, droplet size distribution (DSD(r)), total droplet number concentration, and liquid water content (LWC).

The ShadowGraph system has been effectively utilized in the study of cloud microphysics, both in laboratory settings and during in situ measurements. The droplet detection and sizing mechanisms of the ShadowGraph were comprehensively detailed by (Nowak et al., 2021). Data collected using the ShadowGraph in studies of orographic clouds, specifically under foggy conditions in mountainous regions, were analyzed by (Mohammadi et al., 2022).

During this campaign, the ShadowGraph was used for two purposes: first, as the reference instrument to which the OPC-N3 was calibrated, as demonstrated in Nurowska et al. (2023); and second, to monitor conditions near the surface. The Shadow-Graph operates using a high-power laser with a wavelength invisible to the human eye. For safety reasons, it was installed on the roof of the mobile laboratory, approximately 3 m above ground level.

# 2.2 SolarAOT<sup>upper</sup> station

SolarAOT<sup>upper</sup> - is a private radiative transfer research station (which collaborates with the University of Warsaw) located in an agricultural area on one of the peaks of the Niebylecka Mountain at 445 m a.s.l. (49°52'43.0"N 21°51'40.8"E), located from Strzyżów city in a straight line 4 km, vertical height difference 185 m. The location of both stations is shown on Fig. 1. At the SolarAOT<sup>upper</sup> station are mounted several instruments, inter alia, pyranometer CMP21, Eppley pyrgeometer, CIMEL, Nephelometer Aurora 4000, Aethalometr AE-31, and CHM-15K ceilometer.

**Figure 1.** Location of tethered balloon launching site SolarAOT<sub>lower</sub> and SolarAOT<sup>upper</sup> station, in relation to the Strzyżów city and the Wisłok river.

A Kipp & Zonen CMP21 pyranometer was used to measure downwelling shortwave radiation (285–2800 nm), including both direct solar and diffuse sky components. For longwave radiation, an Eppley pyrgeometer was employed, operating in the spectral range of approximately 4.5 to 50  $\mu$ m, to capture downwelling infrared radiation emitted by the atmosphere and clouds. Both sensors were installed on a leveled platform in an unobstructed area.

CIMEL is an instrument for measuring direct and scattered solar radiation in 9 spectral channels: 340, 380, 440, 500, 675, 870, 936, 1020, 1640 nm. Based on the measured values, the optical parameters of the aerosol are determined, including the AOD and the Ångström exponent. The data collected by the instrument is processed within the international AERONET measurement network. Nephelometer Aurora 4000 is used to measure light scattering coefficients on aerosols for wavelengths of 450, 525, 630 nm in 18 ranges of aerosol scattering angles.

Aethalometer AE-31 is used to measure the concentration of equivalent black carbon (eBC) in the atmosphere and the aerosol absorption coefficient of the aerosol. The measurement is performed at 7 wavelengths (370, 470, 520, 590, 660, 880, 950 nm) using the method of changing the transmission of a quartz filter on which the aerosol is deposited.

# 2.3 Balloon apparatus

Measurements were conducted using two meteorological balloons (for better buoyancy), each approximately 1.5 meters in diameter and filled with helium. The balloons were tethered using the Vaisala TTW111 winch (see Figure 2a). Around two metres below the balloon, the apparatus was mounted on the rope holding the balloon. The apparatus used to mount below the balloon was (see Figure 2b):

- Vaisala radiosonde RS41 collecting data about pressure (p), temperature (T), relative humidity (RH),
- GY-63 MS5611 a high-performance pressure sensor module,

- HYT 939 additional T and RH sensor,
- The Alphasense OPC-N3 an optical particle sensor that measures particle counts across size bins ranging from 0.35 to 40 μm, as well as PM<sub>1.0</sub><sup>1</sup>, PM<sub>2.5</sub> and PM<sub>10</sub>. Here, OPC-N3 was used to gather information about fog droplets based on the article (Nurowska et al., 2023), such as liquid water content (LWC), effective radius r<sub>eff</sub> and N<sub>c</sub>.
  - SENSIRION SPS30 optical PM sensor that measures PM<sub>1.0</sub>, PM<sub>2.5</sub> PM<sub>4</sub>, PM<sub>10</sub> mass concentration
  - TFMini visibility sensor

 AE-51 - miniature aethalometer for measuring the eBC concentration and the aerosol absorption coefficient at a wavelength of 880 nm.

The OPC-N3, an optical particle counter designed by Alphasense Ltd., utilizes a diode laser emitting light at a wavelength of 658 nm, along with an elliptical mirror that directs the laser beam towards a detector. The airflow, driven perpendicularly to the laser beam by an integrated fan, allows for continuous operation. The OPC-N3 quantifies particle number concentration ( $N_c$ ) across 24 size bins, covering a diameter range from 0.35 to 40  $\mu$ m. The onboard algorithm converts  $N_c$  measurements into PM1, PM2.5, and PM10 values, Detailed specification of the OPC-N3 is available in the work by (Hagan and Kroll, 2020).

OPC-N3 devices are considered low-cost sensors, which means that two identical units may not yield consistent results due to device-to-device variability. Therefore, cross-calibration between sensors or calibration against a reference-grade instrument is necessary to ensure measurement accuracy. Additionally, individual OPC-N3 units may exhibit signal drift over time, requiring periodic recalibration to maintain data reliability.

For this reason, it was not possible to directly use the calibration parameters provided in (Nurowska et al., 2023). Instead, the calibration had to be repeated following the methodology described in that work, to ensure compatibility with the specific sensors used in this study. OPC-N3 was calibrated to the ShadowGraph following (Nurowska et al., 2023). Results of  $N_c$ , LWC and  $r_{eff}$  were obtained by taking bins of OPC-N3 measuring particles greater than 1.15  $\mu$ m (bin 7 of OPC-N3).

The calibration equations used between OPC-N3 and Shadowgraph are:

$$LWC_{ShadowGraph} = 6.15 \cdot LWC_{OPC-N3} + 0.11$$
 (1)

$$N_{cShadowGraph} = 4.16 \cdot N_{cOPC-N3} + 32.63 \tag{2}$$

$$\mathbf{r}_{\text{eff}}^{\text{ShadowGraph}} = 0.70 \cdot \mathbf{r}_{\text{eff}}^{\text{OPC-N3}} + 3.81 \tag{3}$$

The OPC-N3 allows calculation of the volume droplet size distribution (vDSD), which can be computed using the formula:

$$vDSD(r_b) = N_b \cdot (\Delta r_b \cdot V_b)^{-1} \cdot r_b^3$$
(4)

<sup>&</sup>lt;sup>1</sup>The number after PM denotes particles with diameters up to the given size in  $\mu$ m which are counted.

(a) Balloon with attached payload and connected to the winch.

(b) Zoom to the balloon payload, showing: inside the box (GY-63, HYT 939), OPC-N3, SPS30, TFMini

Figure 2. Balloon setup.

where  $N_b$  is the number of droplets in a bin,  $V_b$  the volume of a bin,  $\Delta r_b$  is the width of the bin and  $r_b$  is the mean bin droplet radius. Although the obtained vDSD was not calibrated against the ShadowGraph, it provides information on which droplet sizes contribute most significantly to the LWC at a given altitude.

## 210 3 Methodology and model set up

215

## 3.1 Balloon measurements methodology

For three days between 9 - 11 September 2023, the measurements of radiative fog were made in Strzyżów city, Poland. The balloon launch site was located in the valley of Strzyżów city. Four setups were used, as it was not possible due to the buoyancy to mount all instruments at once:

- setup 1: GY-63, HYT 939, OPC-N3, SPS30, TTFMini this setup was most common
  - setup 2: only Vaisala radiosonde RS41
  - setup 3: Vaisala radiosonde RS41, AE-51.
  - setup 4: Vaisala radiosonde RS41, GY-63, HYT 939, OPC-N3, SPS30, TFMini.

Figure 3 shows with colored lines when, during the night, the soundings were done, with colors indicating different setups mounted on the balloon. The same information, but with specific sounding times, can be found in the Appendix A1. In total, 74 soundings were conducted. However, due to data recording issues, 11 soundings lacked complete data and were excluded from further analysis. These are indicated in gray in Figure 3 and Table A1. Soundings were done by unwinding the rope until it started to tilt to the horizon. The balloon was stopped for a few seconds, and the line was wound up. Soundings were done with around 15-minute breaks in between.

**Figure 3.** The figure illustrates the timing of the soundings; different colors represent the specific equipment configurations mounted on the balloon: orange - setup with OPC-N3, blue - setup with radiosonde, pink - setup with OPC-N3 and radiosonde, gray - problems with collected data. The image is overlaid on the line representing temporal variability of the LWC at the ground obtained from Shadowgraph (the same data as on Fig. 5).

| Fog episode |                          | Stage        |             |                                          |  |
|-------------|--------------------------|--------------|-------------|------------------------------------------|--|
|             |                          | Initial      | Developed   | Decaying (soundings after fog vanishing) |  |
| Night 8-9   | Time                     | 23:00 - 2:34 | 2:34 - 6:42 | 6:45 - 7:00 (8:10)                       |  |
|             | Profiles with OPC-N3     | 4            | 12          | 2(+1)                                    |  |
|             | Profiles with Radiosonde | 1            | 6           | 2 (+4)                                   |  |
| Night 9-10  | Time                     | 00:00 - 2:45 | 2:45 - 6:00 | 6:00 - 7:3                               |  |
|             | Profiles with OPC-N3     | 0            | 9           | 2                                        |  |
|             | Profiles with Radiosonde | 1            | 4           | 3                                        |  |
| Night 10-11 | Time                     | 2:00 - 3:02  | 3:02 - 5:30 | 5:30 - 6:00 (8:00)                       |  |
|             | Profiles with OPC-N3     | 1            | 7           | 2 (+1)                                   |  |
|             | Profiles with Radiosonde | 1            | 2           | 0 (+5)                                   |  |

**Table 1.** Times of initial, developed and decaying stage of observed fogs on days 9 - 11 Sep., with information on how many soundings were performed in each period.

The fog case description was divided into 3 stages: initial, developed, and decaying. The transition from the initial to the developed stage was assumed to occur when LWP>15 g·m $^{-2}$ , the change from mature to decaying was assumed when LWP

$$LWC(z) = \int_{z'=0}^{z'=z} \alpha(z')\Gamma_{ad}(T(z'), p(z')) dz' + LWC_0 \approx \Gamma_{ad}(T_B, p_B) \int_{z'=0}^{z'=z} \alpha(z') dz' + LWC_0$$
 (6)

LWP is defined as:

$$LWP = \int_{z'=0}^{z'=CTH} LWC(z') dz'$$
(7)

as fog base is at ground, the integration takes place from z' equal zero to cloud/fog top height (CTH).

In the case of shallow clouds  $\Gamma_{ad}(T(z),p(z))$  can be assumed constant with height Brenguier (1991)  $\Gamma_{ad}(T_B,p_B)=const.$ 250 where  $T_B$  and  $p_B$  are respectively temperature and pressure at fog base/ground. Since the dependence of  $\alpha(z)$  is unknown, the concept of equivalent adiabaticity  $\alpha_{eq}=const.$  is introduced. The  $\alpha_{eq}$  is defined as the constant adiabaticity value that would give the same LWP value when replacing  $\alpha(z')$  in Eq. 6 and calculating LWP from Eq. (7). After taking  $\alpha_{eq}=const.$  the formula for LWP becomes:

$$LWP = \frac{1}{2}\alpha_{eq}\Gamma_{ad}(T_B, p_B) \cdot CTH^2 + LWC_0 \cdot CTH$$
(8)

The formula for LWC with the above assumptions is:

$$LWC(z) = \alpha_{eq} \Gamma_{ad}(T_B, p_B) \cdot z + LWC_0 \tag{9}$$

Figure 4. Representation of profile of T and LWC with added lines of  $\Gamma_{wa}$  and  $\Gamma_{ad}$  respectively. The  $\gamma_{fit}$  and  $\alpha_{fit}$  represents the angle between the best line fit to T and LWC, respectively (from bottom to height of max LWC) and  $\Gamma_{wa}$  and  $\Gamma_{ad}$ .  $\alpha_{eq} = const$ . is defined as deviation from  $\Gamma_{ad}$ , which would give the same LWP as the original data.

The method of calculating  $\Gamma_{ad}(T_B, p_B)$  was taken from Appendix A of the article (Toledo et al., 2021).

To calculate what  $\alpha_{eq}$ , just reverse the Equation 8

$$\alpha_{eq} = \frac{2(\text{LWP} - \text{LWC}_0 \cdot CTH)}{\Gamma_{ad}(T_B, p_B) \cdot CTH^2}$$
(10)

In the literature, instead of  $\alpha_{eq}$ , the parameter  $\beta$  is sometimes used, introduced by Betts (1982) as the in-cloud mixing parameter. This parameter measures departure from the adiabatic situation. The relation between  $\alpha_{eq}$  and  $\beta$  is  $\alpha_{eq} = 1 - \beta$ .

In the latter part of this article will be used:

- $\Gamma_{ad}$  adiabatic condensation rate of LWC,
- $\alpha_{eq}$  scaling of  $\Gamma_{ad}$ , which would give the same LWP for the whole cloud/fog,
- $\alpha_{fit}$  scaling of  $\Gamma_{ad}$  obtained by fitting line to LWC dependence from height.
- $\Gamma_{wa}$  moist adiabatic lapse rate for T,
- $\gamma_{fit}$  scaling of  $\Gamma_{wa}$  obtained by fitting line to T dependence from height.

The Fig. 4 presents the visualization of the concepts listed above.

# 3.3 Radiometer Data Processing


At the SolarAOT<sup>upper</sup> station, a CMP21 pyranometer and an Eppley pyrgeometer were installed to measure downwelling shortwave and longwave radiation, respectively. Both instruments recorded data at 42-second intervals. At the SolarAOT<sub>lower</sub> station, a CNR4 net radiometer was mounted to measure both upward and downward shortwave and longwave fluxes, with a sampling interval of 36 seconds. Data from SolarAOT<sup>upper</sup> were interpolated to match the temporal resolution of the lower station. Short spikes in the radiometric signal—likely due to transient obstructions such as birds—were removed using a filtering algorithm.

Additionally, the signal was smoothed using a 10-minute running mean. The 10-minute averaging window was chosen to correspond to the typical duration of balloon flights (10-15 minutes), allowing for comparison between the measured and simulated radiative fluxes.

## 3.4 1D Simulations radiation fluxes





This section presents the model used to perform optical, microphysical, and radiative closure, as discussed in Section 4.4.

The model was used to assess the consistency between observed radiation and retrieved fog microphysical properties. For this purpose, simulations were done in 1D using the Fu-Liou code (Fu and Liou, 1992, 1993).

The Fu-Liou radiative transfer model is a sophisticated tool designed to accurately simulate radiative transfer in the Earth's atmosphere. The Fu-Liou code uses  $\delta$ - two/four - flux approximation, which allows it to efficiently handle the complexities of radiation scattering and absorption by gases, aerosols, and cloud particles. The model covers six shortwave (SW,  $\lambda$  < 4  $\mu$ m) and 12 longwave (LW,  $\lambda \ge 4 \mu$ m) spectral bands, making it well-suited for various atmospheric conditions. The Fu-Liou model provides detailed insights into the interactions between cloud microphysics and radiation. The model vertical levels span from the ground up to 10 km, with a greater density closer to the surface. In the first 100 m, the grid was spaced every 10 m, and from 100 m to 1 km every 100 m. Input into the Fu-Liou model includes profiles of thermodynamic parameters, fog optical and microphysical quantities, aerosol optical properties, and surface reflectance and emissivity.

To perform simulations, the following specific data were provided to the model:

- T and specific humidity profile. The data from the soundings were combined with the sounding from Tarnów (WMO station 12575) more information is given in Appendix A.
- r<sub>eff</sub>: Due to limitations of the radiative transfer model, r<sub>eff</sub> was assumed to be constant with height within the fog layer. It was calculated using data from the OPC-N3, which measures droplet concentrations in 24 size bins. To exclude aerosol particles, only bins 7 to 24 (corresponding to droplet diameters from 1.15 to 20 μm) were used in the calculation.
- Fog height in the model was assumed that the fog starts at the surface and reaches the CTH level. The top of the fog was determined as the point where LWC <  $0.12 \text{ g} \cdot \text{m}^{-3}$ .
- Aerosol optical depth (AOD) measurements from CIMEL at SolarAOT<sup>upper</sup> were taken. To adjust how much the beam is weakened by the vertical distance between the upper and lower site, the value of AOD was added to the extinction coefficient (obtained from Aurora 4000 and AE-31) times the height difference (185 m) between both stations.
- Aerosol single scattering albedo (SSA) Based on AE-31 and Aurora 4000 located at SolarAOT<sup>upper</sup>, the SSA was calculated. The value of SSA at the moment of the balloon sounding was obtained by linear interpolation.
- Aerosol Ångström Exponent (AE) at 440/870 nm. A CIMEL sun photometer is installed at the SolarAOT<sup>upper</sup> site. For
  the simulations, AE values were rounded to ensure consistent conditions across all cases. Data from both CIMEL and

lidar indicate an influx of Saharan dust during the period of 8–10 September. The approximate AE value recorded by CIMEL during these days was 0.5, increasing to 1.0 on 11 September.

- The asymmetry parameter was derived using Mie scattering theory. Initially, the liquid water content and effective droplet radius were employed to estimate the droplet number concentration, assuming a monodisperse size distribution. Subsequently, spectral optical properties—extinction, scattering, and single scattering albedo—were computed across relevant wavelengths. Finally, the asymmetry parameter was calculated by integrating the angular scattering phase function obtained from classical Mie theory.
- The model allows for the specification of surface albedo based on the International Geosphere-Biosphere Program (IGBP) land cover classification, using one of 20 predefined surface types. For all simulations performed in this study, the IGBP class was set to "grassland" (IGBP = 10), as the measurement site was located on a valley slope predominantly covered with grass, with sparse one-family houses. Surface albedo was implemented as a spectrally resolved, solar zenith, and water vapor content dependent parameter.

## 4 Case study: Valley of Strzyżów city




Fog was observed during three successive nights between 8 and 11 September 2023 in the valley of Strzyżów city. The balloon was launched after fog was visible at the lower station. Fig. 5 presents the situation at the lower and upper stations during fog occurrence. Fig. A2 presents photographs taken on three consecutive days at 04:00 UTC from the SolarAOT<sup>upper</sup>station, showing the top of the fog layer. During the experiment, the fog was not detected at the upper site. The Table 1 presents the duration of each fog and its division into stages. In this section will be described the evolution of each fog as well as its general pattern.

## 4.1 Meteorological overview

The area of Poland, as well as almost all of Europe, was under the influence of anticyclonic circulation of high pressure from Russia. The pressure on 9 September was constant and was 1019 hPa, from 9 UTC on 10 September it began to slowly drop to reach the value of 1012 hPa on 11 September at 11 UTC. During days 8-10 September 2023, there was an event of Saharan dust over Poland. The AE measured for those days by CIMEL at SolarAOT<sup>upper</sup> station was around 0.5 (for a period of Saharan dust) and 1.0 (for the morning of 11 September). The mean AOD during the dust episode was not very high (0.19 at 500 nm).

From the lidar data (Fig. A1) can be seen that the sky was mostly cloudless. On September 9 in the morning, cirrus clouds were visible. The average wind speed did not exceed 2.5 m·s<sup>-1</sup>. Slow advection of hot air of tropical origin caused an inflow of Saharan mineral dust visible at 2-4 km a.g.l. on lidar data A1. The Fig. 5d shows the aerosol scattering coefficient of light at 525 nm (ASC<sub>525</sub>), for three nights of observations. On the night between 8-9 September 2023 ASC<sub>525</sub> was below 100 Mm<sup>-1</sup> which suggests moderate air quality conditions, just before the onset of fog 21:30-22:30 the values peak to 240 Mm<sup>-1</sup> and after the end of fog, values once again peak exceed a very high level of 500 Mm<sup>-1</sup>. These two peaks are probably due to industrial

activity during inversion conditions and the turning on of the heating systems in houses. The morning peak is coincident with inversion disappearance and the transport of pollution from the bottom of the Strzyżów valley. On the night between 9 and 10 September the  $ASC_{525}$  was descending during the night from 150 to 100  $Mm^{-1}$ , with a peak to 250  $Mm^{-1}$  at 21 UTC. The cleanest conditions, with no evening peak of  $ASC_{525}$ , were on night 11 September with values below 100  $Mm^{-1}$ . At the upper station, always in the evening and at night, the values of  $ASC_{525}$  were below 100  $Mm^{-1}$ . The air in the valley was trapped under the inversion of temperature. The inversion formed at 18:00 at night from days 8 and 9, and around 19:00 at night on day 10, September 2023. The course of T in the valley each day was similar during the day, reaching a maximum of 24-26 °C, and reaching a minimum 12.5-13.5 °C around 5 UTC (Fig. 5). The inversion disappeared around 8:40, 7:40, and 8:10, respectively, for days 9, 10, and 11 September 2023. The RH at SolarAOT<sub>lower</sub> station during fog was reaching 100%. The air at SolarAOT<sup>upper</sup> station was lower (RH=60-90%).

The lower panels of Fig. 5 show the calculated visibility in kilometers. Visibility was derived from ShadowGraph measurements using the Koschmieder formula, under the assumption of monodisperse droplets with a radius equal to the  $r_{eff}$  obtained from ShadowGraph, and LWC also provided by ShadowGraph data. An extinction coefficient of 2 was assumed, corresponding to the geometrical optics regime.

During the nights of 9 and 10 September, visibility decreased sharply around midnight, reaching values as low as 100–200 m. The fog dissipated abruptly around 6:00. In contrast, the fog event on the night of 11 September exhibited a different evolution: intermittent patches of fog began to form around midnight, followed by the development of a more continuous fog layer after 02:00, which persisted until approximately 6:00.

# 4.2 Fog microphysics




Based on OPC-N3 measurements, it was possible to compute LWC and LWP; results for each fog are presented in Fig. 6. Observed fogs were occurring mostly in moderate aerosol conditions, and fog layers were located in the range of the T inversion. The fog top was varying from sounding to sounding, mostly it was 85 m (max. 115 m, see Fig 6).

The Fig. 7, 8 and 9 presents the T and RH with height as well as LWC,  $N_c$  and  $r_{eff}$  for each event of fog. The profiles are shown starting from 2 m above the ground, as values below this height could have been significantly affected by surface influence or local disturbances during balloon launch procedures. For this reason, the lines fitted to the profiles were calculated from 2 m to 80% of the fog height. The level of 80% was chosen according to Cermak and Bendix (2011), that above 80% of the height, the fog layer mixes with the dry air above it, which contributes to the reduction of LWC. It is worth mentioning that at that stage of the year, the sunrise is at 4:00 UTC (local time 6:00). Time is given in UTC; for this period of year, UTC is -2 hours from local time.

## 365 4.2.1 Thin-to-tick transition

In the observed fog events, several criteria were considered to identify a possible transition from thin to thick fog. Following Costabloz et al. (2024), we evaluated: IR net radiation at the surface, temperature, cloud top height, and liquid water path.

**Figure 5.** Temporal variability of weather condition on the ground for 09/10/11 Sep. 2023 at the SolarAOT<sub>lower</sub> site (solid lines) and SolarAOT<sup>upper</sup> station (dotted line). On the panels  $\mathbf{a}$ ), $\mathbf{g}$ ), $\mathbf{m}$ ) is presented LWC form ShadowGraph, for reference when the soundings of the balloon with installed OPC-N3 occurred an overlay of Fig, 6 was added. Panels  $\mathbf{b}$ ),  $\mathbf{h}$ ),  $\mathbf{n}$ ) presents the N<sub>c</sub> of droplets registered by ShadowGraph. Panels  $\mathbf{c}$ ),  $\mathbf{i}$ ), and  $\mathbf{o}$ ) shows  $\mathbf{r}_{\text{eff}}$  obtained from ShadowGraph. Figure  $\mathbf{d}$ ),  $\mathbf{j}$ ) and  $\mathbf{p}$ ) presents the ASC at 525 nm from Aurora 4000; panels  $\mathbf{e}$ ),  $\mathbf{k}$ ),  $\mathbf{r}$ ) T, panels  $\mathbf{f}$ ),  $\mathbf{l}$ ),  $\mathbf{s}$ ) RH and panels  $\mathbf{t}$ ),  $\mathbf{u}$ ),  $\mathbf{w}$ ) visibility obtained from ShadowGraph.

**Figure 6.** The bar chart with green-blue colors indicates the change in time of fog LWC with height for days **a**) 09, **b**) 10, **c**) 11 Sep. 2023. The left axis represents the height, while the right axis corresponds to the total LWP for each balloon sounding (marked by an orange diamond). Blue dashed line indicates the 110 m, orange line indicates LWP equal to 15 g·m<sup>-2</sup>. Those are criteria indicating the transition of fog from thin to thick by (Costabloz et al., 2024)

.

Costabloz et al. (2024) proposed five conditions to characterize this transition: (1) longwave net radiation approaches zero, (2) the temperature gradient between 50 m and 25 m becomes negative, (3) TKE exceeds 0.10 m<sup>2</sup> s<sup>-2</sup>, (4) CTH exceeds 110 m and (5) LWP exceeds 15 g m<sup>-2</sup>. They demonstrated that while all conditions were met for thick fog, the exact time when each was fulfilled could differ by up to one hour, making precise estimation of the transition time challenging. In this study, we evaluated four out of five conditions proposed by Costabloz et al. (2024) (without the TKE condition). The following list outlines which criteria were met during each observed fog event.

- On the night of 8–9 September, the LWP exceeded 15 g·m<sup>-2</sup> at 01:30. The IR net radiation criterion was satisfied for three profiles at 02:38, 03:48, and 04:16. The profile at 04:16 satisfied three out of the four criteria (IR net radiation, CTH, and LWP) and was conducted during a brief period of thicker fog just before sunrise. However, this state of thick fog did not persist for long.



- On the night of 9- 10 September, the LWP exceeded 15  $g \cdot m^{-2}$  from the start of valid measurements at 02:30 until the fog dissipated.
- On the night of 10–11 September, the LWP exceeded 15 g·m $^{-2}$  at 03:10, but the CTH did not reach the 110 m threshold.

Although the 15 g·m<sup>-2</sup> threshold suggested by Costabloz et al. (2024) was frequently exceeded, our results indicate that this value alone is insufficient to reliably distinguish between thin and thick fog. Therefore, we interpret the fog event on 9 September as a case in which a transition towards thick fog had started but was interrupted by sunrise before full development. In the fog events on 10 and 11 September, the other criteria were not met. In none of these cases did the LWP exceed 30 g·m<sup>-2</sup>

— the threshold for thick fog proposed by Wærsted et al. (2017). Given that the period of thick fog was very short-lived and most observations remained within the thin fog regime, our findings should be considered primarily representative of optically thin fog conditions.

# 4.2.2 Night 08-09 September 2023






The fog event during the night of 8- 9 September 2023 was observed from its development stage (23:00–01:41), through its mature stage (02:45–06:42), until dissipation (06:45–07:00). Fig. 7 presents vertical profiles of microphysical parameters, including LWC,  $N_c$  and  $r_{eff}$ , along with T and RH. The equivalent adiabaticity ( $\alpha_{eq}$ ) is shown in the figures displaying LWC profiles. The division into fog life cycle stages is illustrated in Fig. 7 by pink, blue, and yellow ochre corresponding to the development, mature, and dissipation stages, respectively. Apart from that, separate figures can be found in the Appendix for each stage: Fig. A3a, Fig. A3b, and Fig. A3c. Each of the fog stages is described below.

- **Development of fog:** There were 5 soundings conducted between 23:00–2:34 (see Fig. A3a). The T was decreasing with height from the ground to 40 m a.g.l., starting from T = 12.0 °C and decreasing at the rate  $\gamma_{fit} \cdot \Gamma_{wa} = 0.48 \cdot (-5.0)$ °C·km<sup>-1</sup>. Above 40 m, a temperature inversion was present, with T reaching 14.7 °C at 100 m. The top of the fog was 65 m. The RH was constant, equal to 100% up to 87 m, and in the last 10 m it dropped with height (96% at 100 m). LWC was slightly increasing with height ( $\alpha_{fit} \cdot \Gamma_{ad} = 1.34 \cdot 2.32 \text{ g·m}^{-3} \cdot \text{km}^{-1}$ ), from 0.18 to a maximum of 0.30 g·m<sup>-3</sup> at 23 m a.g.l.; up to 36 m a.g.l., LWC was oscillating near 0.26 g·m<sup>-3</sup>, and above that it decreased. The  $\alpha_{eq}$  was 0.29. The LWC<sub>ShadowGraph</sub> (referring to LWC measured by the ShadowGraph, Fig. 5) shows that during the development stage, LWC<sub>ShadowGraph</sub> increased, reaching its maximum of 0.46 g·m<sup>-3</sup> at 1:47. Values of LWC from ShadowGraph were higher than those from OPC-N3. The LWP (Fig. 6) increased from 5.63 g·m<sup>-2</sup> at 00:25 to 12.97 g·m<sup>-2</sup> at 1:36. Mean N<sub>c</sub>increased with height to 247 cm<sup>-3</sup> at 35 m, then decreased with height to 48 cm<sup>-3</sup> at 65 m. The r<sub>eff</sub>decreased steadily with height from 11.2 to 5.7 μm at the CTH.
- Mature state of fog: There were 13 soundings conducted from 2:34 till 6:42 (Fig. A3b). The T and RH showed a similar pattern with height as in the previous stage. The fog was deeper, with a top at 102 m. However, the temperature inversion started higher, around 60 m above the ground, and the lapse rate in the lower part was higher:  $\gamma_{fit} \cdot \Gamma_{wa} = 1.16 \cdot (-5.1)^{\circ} \text{C·km}^{-1}$ . The  $N_c$  maximum, equal to 410 cm<sup>-3</sup>, was at 48 m; above this,  $N_c$  decreased to 65 cm<sup>-3</sup> at 90 m. Above that, the number of drops was constant. The  $r_{eff}$  profile changed and can be divided into two sections. From the ground to a height of 88 m,  $r_{eff}$  remained almost constant (at bottom 9.2  $\mu$ m; 8.3  $\mu$ m at 88 m). From 88 m to CTH,  $r_{eff}$  decreased sharply with height (mean at the top 5.2  $\mu$ m). Because the  $N_c$  maximum is shifted upward, the LWC maximum is also at a higher altitude (56 m). The  $\alpha_{fit}$  is positive, equal to 0.90, and  $\alpha_{eq}$  is equal to 0.30. The images from the ShadowGraph indicate that LWC<sub>ShadowGraph</sub> near the ground decreased over time starting from 2:57, and this was associated with a decrease in  $r_{eff}$  and not  $N_c$ .
- **Disappearing stage:** There were 2 soundings conducted between 6:45 and 7:00 (Fig. A3c). The T increased from the ground to 27 m, above which T decreased with height. Unfortunately, the sounding with the Vaisala radiosonde

RS41 was interrupted at 45 m. Between the two soundings, spaced less than 15 minutes apart, the T profile shifted by +2 °C. At this time, the RH profile dropped by 5%. In the first 20 m, the mean RH dropped from 96 % near the ground to 91% at 20 m. The fog evaporated quickly. The  $\alpha_{eq}$  was positive, equal to 0.33. LWC increased with height  $(\alpha_{fit} \cdot \Gamma_{ad} = 0.42 \cdot 2.39 \, \text{g·m}^{-3} \cdot \text{km}^{-1})$ , reaching a maximum at 72 m (mean LWC 0.26 g·m<sup>-3</sup>); at almost the same height,  $N_c$  also reached its maximum (488 cm<sup>-3</sup> at 74 m). The layer above was characterized by a rapid decrease in both values up to the CTH. The  $r_{eff}$  was constant with height up to 80 m, around 6.8  $\mu$ m, except for the layer from the ground to 18 m, where  $r_{eff}$  was higher, up to 9.5  $\mu$ m.

The Fig. 7 presents the mean values with height for the whole-night fog event from 08-09 September 2023. For the entire fog event,  $\alpha_{eq}$  is 0.23. As a first approximation, fog  $r_{eff}$  decreases linearly with height, while  $N_c$  can be approximated by a quadratic equation. The equations for fitted lines are respectively:

$$\mathbf{r}_{\text{eff}} = 0.03 \cdot h + 9.06 \ [\mu \text{m}],$$
 (11)

$$N_c = -0.10 \cdot h^2 \cdot 10 + 8.27 \cdot h + 130.24 \text{ [cm}^{-3]}.$$
 (12)

# 4.2.3 Night 09-10 September 2023

420

435

Below are described the stages of fog from 09-10 September 2023:

- **Development of fog:** Due to a malfunction of the apparatus, the development stage of fog with microphysics measurements in the vertical direction was not captured. The fog started at 00:00; however, it is not possible to determine when this stage ended. The first OPC-N3 sounding was registered at 2:34, with LWP > 15 g·m<sup>-2</sup>.
  - Based on ShadowGraph measurements, LWC and  $N_c$  increased continuously until 00:37, reaching local maxima: LWC<sub>ShadowGraph</sub> = 0.34 g·m<sup>-3</sup> and  $N_c$  = 271 cm<sup>-3</sup>. The r<sub>eff</sub> reached its local maximum later, at 1:17, equal to 11.6  $\mu$ m. After the peak, values of LWC,  $N_c$ , and r<sub>eff</sub> fluctuated, reaching their global maxima ( $N_c$  = 388 cm<sup>-3</sup> at 2:18) and minima ( $r_{eff}$  = 7.5  $\mu$ m at 2:28).
- Two profiles of T and RH are shown in Fig. A4a. The profile reaching a higher altitude was performed earlier, at 00:53, and the second about one hour later. The temperature profile was nearly constant in the first 40 m (12.65 °C at the ground), above which a temperature inversion was present that weakened with time. The RH was 100% up to 67 m, then decreased with height. One hour later, RH was constant at 100% throughout the entire column from the ground up to 86 m.
- Mature state of fog: There were 13 soundings conducted from 2:45 to 6:42 (Fig. A4b). The fog height was 87 m. The temperature decreased with height up to 50 m a.g.l., above which a temperature inversion occurred. From the ground to the CTH, RH remained above 99.5%. The maximum LWC was observed at 53 m, with a value of 0.40 g⋅m<sup>-3</sup>.

Figure 7. Vertical profiles of specific quantities measured by the balloon for night 08-09 Sep. 2023. From left: T and RH from Vaisala radiosonde RS41, LWC,  $N_c$ ,  $r_{eff}$  within the fog from OPC-N3. Each colored line represents an individual balloon profile, with different colors indicating different stages of fog evolution: pink corresponds to the formation stage, blue to the mature stage, and yellow ochre to the dissipation stage. The black thick line represents the mean of all the soundings, the colored area represents the range between +/- standard deviation from the mean. At the T plot dotted line presents the wet adiabatic lapse rate  $\Gamma_w$ , dashed red line presents linear fit of T from 2 m to height of maximum mean LWC.  $\gamma_{fit}$  is a scaling factor of  $\Gamma_w$  to obtain the equation of linear fit. At the LWC plot dotted line presents the LWC adiabatic lapse rate  $\Gamma_{ad}$ , the dashed red line presents the linear fit to LWC from 2 m to the height of maximum mean LWC. Where  $\alpha_{fit}$  is a scaling factor of  $\Gamma_{ad}$  obtained by fitting line to LWC dependence from height and  $\alpha_{eq}$  is a scaling of  $\Gamma_{ad}$  which would give the same LWP for the whole cloud/fog. On the  $N_c$  plot, the yellow line indicates the quadratic fit to the data (from 2 m to 80% height of CTH). On the  $r_{eff}$  plot the yellow line indicates the linear fit to the data (from 2 m to 80% height of CTH).

**Figure 8.** Vertical profiles of specific quantities measured by the balloon for night 09-10 September 2023. The detailed description is given in the caption of Fig. 7.

450

A linear fit to the LWC profile from 2 m to 53 m yielded a growth rate equal to 0.51 of the LWC adiabatic lapse rate. The maximum  $N_c$  was lower than during the previous night, reaching 345 cm<sup>-3</sup>. The  $r_{eff}$  at the ground was higher than in the previous day's fog; however, it decreased with height in the first 30 m, remained approximately constant (9.0  $\mu$ m) from 30 to 63 m, and then decreased again toward the CTH.

The LWP (Fig. 6) oscillated between 18 and 23  $g \cdot m^{-2}$ . Most of the water was located in the upper part of the fog, between 30 and 70 m.

- Disappearing stage: Five soundings were conducted between 6:30 and 7:30 (Fig. A4c). The temperature at 2 m a.g.l. and throughout the column increased rapidly (from 11.9 °C at 6:30 to 16.22 °C at 7:30). In the first 54 m, T decreased with height, and above this level, a temperature inversion was observed. As the sun rose, RH decreased from 100% to 88% at 2 m a.g.l.

The parameters  $\alpha_{fit}$  and  $\alpha_{eq}$  were 0.30 and 0.24, respectively. LWC values were below 0.12 g·m<sup>-3</sup> in the first 21 m above the ground. The maximum LWC (0.15 g·m<sup>-3</sup>) occurred at 43 m. When fog almost dissipated (6:23; Fig. 6), remaining fog patches were still observed between 30 and 50 m, with LWC > 0.15 g·m<sup>-3</sup>. The fog top dropped to 57 m.

Fog droplet diameter decreased with height from 6.7  $\mu$ m at 4 m a.g.l. to 4.8  $\mu$ m at the CTH. N<sub>c</sub> fluctuated around 170 cm<sup>-3</sup> between 24 m and 56 m. The fog dissipated from both the top and bottom.

The temperature in the first 53 m was almost constant with height. Within this layer, LWC increased with height at a rate of  $\alpha_{fit} \cdot \Gamma_{ad} = 0.44 \cdot 2.4 \text{ g} \cdot \text{m}^{-3} \cdot \text{km}^{-1}$ , reaching a local maximum of LWC = 0.33 g·m<sup>-3</sup> at 53 m. The corresponding value of  $\alpha_{eq}$  was nearly zero (0.01).

Fig. 8 summarizes the microphysical properties of the fog event on 09-10 September 2023. The following curves were fitted to the values of  $N_c$  and  $r_{eff}$ :

$$N_c = -0.10 \cdot h^2 \cdot 10 + 7.97 \cdot h + 118.98 \,[\text{cm}^{-3}],\tag{13}$$

470 
$$r_{\text{eff}} = 4.29 \cdot h \cdot 10^{-2} + 9.47 \, [\mu \text{m}],$$
 (14)

## 4.2.4 Night 10-11 September 2023




The fog pattern on the night of 10-11 September looks distinct from previous nights. The fog could not form until 3:08, when it started to develop with an abrupt jump in LWC from 0.05 at 3:08 to 0.30 at 3:31. The maximum peak in LWC observed on the ground by ShadowGraph was at 4:27, equal to 0.48 g·m<sup>-3</sup>. Fog rapidly intensified, reaching high LWC values (mean 0.48 g·m<sup>-3</sup>) in the fog body from 10-50 m, with maximum LWC 0.97 g·m<sup>-3</sup> at 31 m at 3:23. At 4:40, high values of LWC above 0.40 g·m<sup>-3</sup> were distributed in the range from the fog bottom to 80 m. As quickly as it appeared, the fog dissipated by 5:40. However, as before, the fog was dissipating more from the bottom than from the top.

During the night of 10-11 September 2023, all stages of fog were captured; each stage is described in detail below. This fog event developed later in the night than previous cases and exhibited more abrupt behavior.

- Development of fog: Between midnight and 2:28, the ShadowGraph was detecting droplets; however, the LWC<sub>ShadowGraph</sub> was below 0.1 g⋅m<sup>-3</sup>. Two soundings were performed, one with OPC-N3 and one with the Vaisala radiosonde RS41, between 2:00 and 3:02 (Fig. A5a). The profile from 2:11 shows that the fog was just forming. LWP was 5.23 g⋅m<sup>-2</sup>.

Fog was confined to the first 23 m in height. Even though the fog was shallow, it had high LWC values at some levels (max. LWC was 0.67 g·m<sup>-3</sup> at 13 m). The  $\alpha_{eq}=-3.22$ , however  $\alpha_{fit}=-0.15$ .

**Figure 9.** Vertical profiles of specific quantities measured by the balloon for the night of 10-11 September 2023. A detailed description is given in the caption of Fig. 7.



The fog was dense—maximum  $N_c$  was 416 cm<sup>-3</sup> at 18 m. The  $r_{eff}$  was decreasing with height (9.8  $\mu$ m at 2 m and 6.0  $\mu$ m at the CTH). The profile from the Vaisala radiosonde RS41 at 2:43 shows that T remained almost constant in the first 40 m (around 12.7-12.8 °C), later slightly increasing with height to 14.2 °C at 100 m.

The RH profile was constant with height; however, the air was not fully saturated (RH  $\approx$  98.5%). ShadowGraph shows that LWC dropped to 0 at 2:38, and within the next hour, rapidly rebuilt to 0.30 g·m<sup>-3</sup>. This was correlated with rapid growth of  $r_{eff}$  from 7.8  $\mu$ m to 12.8  $\mu$ m, while  $N_c$  remained low (2-65 cm<sup>-3</sup>).

- Mature state of fog: Between 3:10 and 5:30, eight soundings with a balloon (Fig. A5b) were conducted. The fog deepened to 83 m. The profiles of T and RH changed: only in the first 10 m was RH above 99.5%, above which it decreased to 85.5% at 60 m.

Unfortunately, the sounding with the Vaisala radiosonde RS41 did not reach the CTH. There was a strong inversion; the fitted lapse rate was  $\gamma_{fit} \cdot \Gamma_{wa} = -11.82 \cdot -4.97^{\circ} \text{C} \cdot \text{km}^{-1}$ . T increased from 12.3 °C at the ground to 15.8 °C at 60 m.

The N<sub>c</sub> profile had a different form compared to previous fog events: it exhibited two protrusions with maxima at 25 m (377 cm<sup>-3</sup>) and 70 m (257 cm<sup>-3</sup>), and a local minimum at 50 m. The  $r_{eff}$  slightly decreased from the ground (10.0  $\mu$ m) to 60 m (8.8  $\mu$ m), then decreased more sharply to 5.5  $\mu$ m at the CTH.

The LWC profile showed a more intermittent pattern, with values fluctuating between 0.2 and 1.2 g·m<sup>-3</sup> and a peak of 0.67 g·m<sup>-3</sup> at 27 m. The  $\alpha_{eq} = 0.56$ , and the fitted  $\alpha_{fit} = 4.80$ . The LWP was the highest of all three events; for four soundings, LWP exceeded 26.5 g·m<sup>-2</sup> (maximum: 27.36 g·m<sup>-2</sup> at 4:27).

- Disappearing stage: The final stage of fog was observed from 5:30 to 6:00. Two soundings were conducted with OPC-N3. In Fig. A5c, two additional soundings of T and RH between 6:00 and 6:33 are also shown. The CTH was at 79 m.
LWC increased with height, with a maximum near the CTH (max. LWC = 0.19 g·m<sup>-3</sup> at 75 m). Due to the location of the maximum LWC near the CTH, α<sub>eq</sub> = 0.32, which was similar to α<sub>fit</sub> = 0.30.

 $N_c$  increased with height to 27 m (maximum 190 cm<sup>-3</sup>), then oscillated around 145 cm<sup>-3</sup> up to 43 m, and sharply decreased to 62 cm<sup>-3</sup> at 48 m. The r<sub>eff</sub> slightly decreased with height, with fluctuations around 6  $\mu$ m.

Fig. 9 summarizes the microphysical properties of the fog event on 10-11 September 2023. The  $\alpha_{eq}$  for the entire event was 0.24. The following curves were fitted to the values of  $N_c$  and  $r_{eff}$ :

$$N_c = -0.04 \cdot h^2 + 1.47 \cdot h + 219.65 \text{ [cm}^{-3]},$$
 (15)

$$r_{\text{eff}} = 4.61 \cdot h \cdot 10^{-2} + 9.57 \,[\mu\text{m}].$$
 (16)

# 4.3 Evolution of fog droplet spectrum




From the OPC-N3 measurements, it was possible to compute vDSD(r) presented in the Fig. 11 a). The vDSD is presented from bins of radius from 1.15 to 20  $\mu$ m to remove aerosol particles. Near the ground was located ShadowGraph, Figure 10 presents the comparison between the vDSD obtained from ShadowGraph and OPC-N3. ShadowGraph shows that near the ground, there are droplets of radius greater than 20  $\mu$ m, and that in the case of OPC-N3, those droplets are counted in the last bin. As it was stated by Nurowska et al. (2023), even though the manufacturer declares that the upper limit of the last bin is 20  $\mu$ m, in fact, the last bin also counts larger particles.

Although OPC-N3 is not calibrated to match the vDSD values, it has a similar pattern of spectrum as ShadowGraph. The vertical profile of the vDSD (Fig. 11a) provides information on which droplet sizes contribute most to the LWC at a given height.

**Figure 10.** vDSD near ground for mature stage of night events of fog on: I. 08-09 Sep., II. 09 - 10 Sep., III. 10-11 Sep. 2023. Panels **a**) and **b**) presents the vDSD obtained from Shadowgraph, while **c**) presents vDSD from OPC-N3. The x axis represents the edges of the bins of droplet radii measured by OPC-N3, plus additionally greater bins (above  $20 \mu m$ ) visible only by Shadowgraph. The panels **b**) presents the same vDSD as panel **a**) however all the droplets with radius greater than  $20 \mu m$  are counted as part of the last bin of OPC-N3 (18.5-20  $\mu m$ ) - this is done to be able to compare the vDSD from OPC-N3 and Shadowgraph. The imaging area for a given device is marked in white.

**Figure 11.** Vertical profile of vDSD and normalized vDSD for 9-11 Sep. 2023 fog occurrence. Panel **a**) presents the vDSD. The scale is divided into steps of  $100 \, \mu \text{m}^2 \cdot \text{cm}^{-3}$  from 0 to  $1000 \, \mu \text{m}^2 \cdot \text{cm}^{-3}$  and then in steps of  $500 \, \mu \text{m}^2 \cdot \text{cm}^{-3}$ . Panel **b**) presents the normalized at each height vDSD. The figure presents what percentage of the entire spectrum at a given height is contributed by the volume of drops from a given bin.

Figure 11**b** shows the normalized vDSD, obtained by dividing the droplet size distribution at each height by  $\sum_{r_b} \text{vDSD}(r_b)$  at that height. This normalization highlights the relative contribution of each size bin to the vDSD at a given altitude.

Figure 11a indicates the altitudes where most LWC is produced and by which droplet sizes, while Fig. 11b enables analysis of the droplet spectrum in regions with low LWC — such as near the top of the fog layer or during dissipation. Apart from vDSD for the whole episode, in the Appendix are shown vDSD (Fig. A6) for each stage of fog: beginning, mature, and disappearing. This section describes how LWC, LWP, and the droplet spectrum evolve during fog occurrence for each night case.



From vDSD (Fig. 11) it is visible that most of the LWC is associated with two drop radius regions. The first region is described by an asymmetric distribution. The maximum value of the distribution is associated with a radius 4-5  $\mu$ m. The distribution has a bigger slope on the left side (droplets smaller than the maximum). The second region is a peak for droplets of radius bigger than 18.5  $\mu$ m ( $r_{>18.5}$ ). Big droplets are found in the whole range of altitudes; however, there are more of them when closer to the ground.

In the 10 m layer closest to the ground, droplets with  $r>18.5~\mu{\rm m}$  contribute up to 40% of the total LWC. Within this layer, the closer to the surface, the smaller the contribution to LWC from droplets with  $r<7~\mu{\rm m}$ , and the larger the contribution from droplets with  $r>7~\mu{\rm m}$ .

Above 10 m, the situation is reversed: droplets with  $r < 7~\mu \text{m}$  contribute more to LWC than those with  $7~\mu \text{m} < r < 18.5~\mu \text{m}$ . Above 40 m, with increasing altitude, the 4-5  $\mu \text{m}$  peak in the normalized vDSD gradually shifts toward larger droplet sizes, reaching approximately 8–9  $\mu \text{m}$ .

The maximum CTH during fog on 9 September was approximately 102 m. In most cases, the LWC above 80 m was below  $0.2 \text{ g} \cdot \text{m}^{-3}$ , indicating only sparse droplets in this region. Just above the CTH, most of the water was accumulated in droplets with radii between 8 and  $14\mu\text{m}$ . With increasing height, this droplet size range decreased.

Subsequent fog nights had increasingly larger LWC at a specific height (see Fig. 6). Even though LWC was reaching higher values on 10 September than on 9 September. The LWP was higher on 9 September because the fog was reaching higher altitudes. The increase of LWC was related to the appearance of droplets in the size of 7-17  $\mu$ m, and not to the increase in the number of droplets in the size of 4-5  $\mu$ m.

In the Appendix the Fig. A6 presents vDSD for three stages of fog for each day. Even in the initial stage of the fog, there were already large drops with  $r_{>18.5}$ , and the fog started to grow in thickness from the bottom. In the case of the fog from the nights of 9 and 10 September, with increasing height, water was stored by drops with increasingly larger radii between 2-10  $\mu$ m. In the case of the fog from the night of 10 to 11 September, with increasing height, an inverse relationship occurs increasingly smaller drops store the most water from the range of radii 2.0-18.5  $\mu$ m.

In the dissipating stage of fog on 09 and 11 September, the CTH did not decrease; it remained around 80 m. While below 30 m, the vDSD shows minimal signal (bottom panels of Fig. A6), suggesting a very low droplet concentration in this region. This suggests that the fog disappeared more from the bottom than from the top.

As the profile of the dissipating fog on 10 September was taken approximately half an hour after the dissipation began, it captured only small droplets in the range of 2–7  $\mu$ m at heights between 20 and 60 m. In all cases, large droplets ( $r_{>18.5}$ ) ceased to contribute significantly to the LWC during the fog dissipation phase.

## 4.4 Optical, microphysical and radiation closure





A radiative closure was performed to assess the consistency between observed radiative fluxes and the microphysical measurements. Given that the OPC-N3 is a low-cost optical particle counter, we wanted to verify whether the vertical profiles of microphysical parameters were representative of actual conditions. Since no independent method was available, we used the obtained microphysical parameters as input for radiative transfer simulations. These simulations enabled us to test whether the microphysical measurements from the OPC-N3 are consistent with the observed radiative fluxes.

Radiative transfer simulations were conducted using the Fu-Liou model in 1D mode, incorporating detailed vertical profiles of thermodynamic and microphysical properties. The model covers six shortwave and twelve longwave spectral bands, with input data including fog microphysics, aerosol optical properties, and surface reflectance. The fog droplet asymmetry parameter

Figure 12. Scatter plots comparing observed and modeled SW (first and third column from left) and LW fluxes under clear-sky (left two columns) and fog conditions (right two columns). The last column presents the difference ( $\Delta$ ) between upper and lower station measurements for observed and modeled LW flux in fog conditions. The red solid line indicates the linear fit to the data. Blue triangles represent measurements from the night of 8–9 Sep., black circles show fog data from the night of 9–10 Sep., and pink triangles correspond to data collected during the night of 10–11 Sep. 2023. The equation for each fit is shown in the corresponding panel.

was calculated using Mie theory, based on measurements of liquid water content and droplet size. Details of the model setup are provided in Section 3.4.

For performing the simulations, only cases when setup 1 (with OPC-N3) was attached to the balloon and data were properly collected were used. The simulated shortwave (SW) and longwave (LW) fluxes were compared with fluxes registered at the SolarAOT<sup>upper</sup> and SolarAOT<sub>lower</sub> stations. Two flights on 11 September between 05:40 and 06:22 were excluded from the radiative closure analysis due to suspected water condensation on the lower LW radiometer during fog dissipation. In total, there were 37 soundings used for analysis.



Fig. 12 compares SW and LW radiation between the model and observations at the SolarAOT<sup>upper</sup> and SolarAOT<sub>lower</sub> stations. The scatter plots also show linear fits to the data. The left panels present the model results for clear-sky conditions based on observations from 10 September, while the right panels present results from the time when there were fog conditions during days 9-11 September.

Upper-air temperature and humidity profiles were taken from balloon soundings by the Polish Meteorological and Water Management Institute in Tarnów, available only twice daily (00 and 12 UTC), resulting in limited temporal resolution above

**Figure 13.** Incoming SW flux for SolarAOT<sup>upper</sup> and SolarAOT<sub>lower</sub> station. The black solid line measured data, the yellow dashed line model results for no fog conditions, the orange circles - model results for fog conditions measured by soundings. The third panel from the top presents the difference in SW flux between the upper and lower site (solid line), pink circles represent the difference between SolarAOT<sup>upper</sup> station and model when fog conditions were implemented. Blue triangles represent the difference between the SolarAOT<sub>lower</sub> station and the model with implemented fog conditions. The lowest panel presents the total net radiation for SolarAOT<sup>upper</sup> (black line) and SolarAOT<sub>lower</sub> station (orange line).

100 m. In contrast, near-surface profiles were measured more frequently on site (when fog was present). Therefore, for comparison of clear-sky conditions, only data from near the sounding at  $12 \pm 2$  hours were used.



The equation describing the relationship between the SW radiation under clear-sky and fog conditions for both stations has a relation of almost 1:1. The offset is up to  $-12 \text{ W} \cdot \text{m}^{-2}$ .

Given the limited variation in LW downward radiation and the uncertainties introduced by sparse temperature soundings, absolute comparisons via regression provide limited insight. Instead of separate scatter plots for two stations, we present the difference in LW radiation between the upper and lower stations. The offset of the linear fit by  $13.10 \text{ W} \cdot \text{m}^{-2}$  suggests that the fog implemented in the model can be too thick.

**Figure 14.** Incoming LW flux for SolarAOT<sup>upper</sup> (upper panel) and SolarAOT<sub>lower</sub> station (middle panel). The black solid line measured data, the yellow dashed line - model result for no fog conditions, the orange circles - model results for fog conditions measured by soundings. Lower panel presents with solid line difference in LW flux between SolarAOT<sup>upper</sup> and SolarAOT<sub>lower</sub> station, pink circles represent difference between upper station and model with fog conditions implemented and blue triangles represent difference between lower station and model with fog conditions.

|                                                      |                  | MBE             | $\mathbf{E}[\mathbf{W}\cdot\mathbf{m}^{-2}]$ |                |             |                |
|------------------------------------------------------|------------------|-----------------|----------------------------------------------|----------------|-------------|----------------|
| Night between 08-09 Sep.                             |                  | )9 Sep.         | 09-10 Sep.                                   |                | 10-11 Sep.  |                |
| Model run                                            | every 5 min      | when sounding   | every 5 min                                  | when sounding  | every 5 min | when sounding  |
| Fog implementation                                   | no               | yes             | no                                           | yes            | no          | yes            |
| Time                                                 | 7-17 UTC         | 4 -7 UTC        | 7-17 UTC                                     | 4 -7 UTC       | 7-17 UTC    | 4 -7 UTC       |
| $I_{Solar AOT^{upper}}$                              | $-12.9 \pm 22.3$ | $12.7 \pm 23.1$ | $-4.5 \pm 2.9$                               | $1.4 \pm 3.3$  | -           | $6.5 \pm 5.8$  |
| $I_{SolarAOT_{lower}}$                               | $-1.0 \pm 23.1$  | $8.2 \pm 23.7$  | $6.7 \pm 8.1$                                | -1.5 ± 15.4    | -           | $-0.3 \pm 8.9$ |
| I <sub>SolarAOT<sub>lower</sub></sub> (-cirrus bias) | $11.9 \pm 13.9$  | $0.8 \pm 26.0$  |                                              |                |             |                |
| Time                                                 | 10-14 UTC        | 0 -7 UTC        | 10-14 UTC                                    | 0 -7 UTC       |             | 0 -7 UTC       |
| IR <sub>SolarAOT</sub> upper                         | -29.9 ± 2.9      | $-3.7 \pm 3.8$  | $-32.4 \pm 2.3$                              | $2.6 \pm 3.4$  | -           | $0.3 \pm 6.0$  |
| $IR_{SolarAOT_{lower}}$                              | $-20.7 \pm 1.5$  | $9.3 \pm 8.6$   | -22.3 ± 3.1                                  | $10.8 \pm 7.9$ | -           | $12.2 \pm 3.3$ |
|                                                      |                  | RMSI            | E [W·m <sup>-2</sup> ]                       |                | ,           |                |
| Night between                                        | 08-09 Sep.       |                 | 09-10 Sep.                                   |                | 10-11 Sep.  |                |
| Model run                                            | every 5 min      | when sounding   | every 5 min                                  | when sounding  | every 5 min | when sounding  |
| Fog implementation                                   | no               | yes             | no                                           | yes            | no          | yes            |
| Time                                                 | 7-17 UTC         | 4 -7 UTC        | 7-17 UTC                                     | 4 -7 UTC       | 7-17 UTC    | 4 -7 UTC       |
| $I_{SolarAOT^{upper}}$                               | 25.7             | 25.4            | 5.4                                          | 3.3            | -           | 8.4            |
| $I_{SolarAOT_{lower}}$                               | 23.1             | 24.0            | 10.5                                         | 14.1           | -           | 8.0            |
| I <sub>SolarAOT<sub>lower</sub></sub> (-cirrus bias) | 18.2             | 24.8            |                                              |                |             |                |
| Time                                                 | 10-14 UTC        | 0 -7 UTC        | 10-14 UTC                                    | 0 -7 UTC       |             | 0 -7 UTC       |
| IR <sub>SolarAOT</sub> upper                         | 30.1             | 5.3             | 32.5                                         | 4.1            | -           | 5.7            |
| $IR_{SolarAOT_{lower}}$                              | 20.8             | 12.5            | 22.5                                         | 13.2           | -           | 12.6           |

**Table 2.** Statistics of SW (I) and LW (IR) flux comparisons between the model and observations at both sites. The mean bias error (MBE) and root mean square error (RMSE) were calculated for each day under fog and non-fog conditions. Additionally, for Sep. 9, the mean MBE and RMSE for the SW radiation were computed after removing the estimated cirrus-induced bias.

Figures 13 and 14 show the temporal comparison of SW and LW fluxes between the model and observations at the SolarAOT<sup>upper</sup> and SolarAOT<sub>lower</sub> stations. Black lines show the measured incoming SW and incoming LW radiation fluxes. By the yellow dashed line is presented the simulation result for clear-sky, while the orange circles present the result of the simulation with implemented fog conditions, based on soundings. The difference between observations and the model is shown in Fig. 14 on the lower panel for LW flux and in Fig. 13 on the second panel from bottom for SW flux. Additionally, the Table 2 presents the statistics between simulated and measured SW and LW flux for clear-sky (no fog) and fog conditions.



For clear-sky SW radiation, the model flux at the SolarAOT<sup>upper</sup> station is underestimated. The root mean square error (RMSE) for 9 and 10 September is 25.7 and  $5.4 \cdot m^{-2}$ , respectively. On 9 September, the SW radiation at the SolarAOT<sup>upper</sup> station exhibits a rugged temporal pattern due to the presence of cirrus clouds. The relatively high RMSE on this day is

attributed to cirrus cloud contamination, which was not accounted for in the radiative transfer model. For the SolarAOT<sub>lower</sub> station, the RMSE is 23.1 and 10.5 W·m<sup>-2</sup>, respectively, for 9 and 10 September.

The model LW flux at both sites for clear-sky conditions is underestimated, and the RMSE exceeds  $30 \text{ W} \cdot \text{m}^{-2}$  for the upper station. Running the model based solely on aerological soundings performed twice daily does not allow for an accurate representation of the diurnal variation of LW radiation. The results show significant discrepancies between the model and observations at both the upper and lower stations. To better reproduce the temporal evolution of IR radiation, more detailed information on the distribution of RH in the lower atmospheric layer is required.

The results indicate that when fog conditions are included, the model reproduces both SW and LW fluxes reasonably well at both stations. This is due to inputting more information from our soundings. For the SolarAOT<sup>upper</sup> station, SW fluxes are slightly overestimated (by up to 12.7 W·m<sup>-2</sup>), while LW fluxes show deviations within  $\pm 4 \text{ W·m}^{-2}$ .

At the SolarAOT<sub>lower</sub> station, biases for both SW and LW fluxes remain within  $12.2~W\cdot m^{-2}$ . The RMSE generally stays below  $14.2~W\cdot m^{-2}$ , except under cirrus cloud conditions (9 September). The RMSE values for LW radiation at the lower station are approximately  $12-13.5~W\cdot m^{-2}$  during fog conditions, indicating that the chosen fog microphysical parameters — liquid water content LWP and  $r_{eff}$  — adequately represent the fog's thermal radiative properties. The relatively low deviation suggests that the modeled fog layer produces realistic LW fluxes near the surface, supporting the suitability of the microphysical assumptions for the observed conditions. Due to model simplifications, the fog is assumed to have a constant droplet size and LWC at all heights, which may introduce some uncertainty into the results. As can be seen from the temporal comparison of LW radiation from model and data (Fig. 14) most of the model overestimation occurs during the fog decay stage; these errors may result from the lack of homogeneity, the patchwork nature of the fog during its decay. Overall, these results demonstrate that the model captures the radiative fluxes with acceptable accuracy during fog events.

# 4.5 Radiative Fluxes During Fog Events





The apparatus at SolarAOT<sup>upper</sup> and SolarAOT<sub>lower</sub> station measures the total net radiation (NET; downward minus upward SW+LW fluxes), which is presented on the lowest panel of Fig. 13. During the first night of observations, it is visible that between 00:00 and 00:44 the NET radiation at the SolarAOT<sub>lower</sub> station changed from -24.4 to -6 W·m<sup>-2</sup>. After the development of fog, the NET radiation at the lower station was around 0 W·m<sup>-2</sup>. It became positive after sunrise. The difference between lower and upper station NET radiation during night fog was around 50 W·m<sup>-2</sup>. When the fog disappeared (7:00), there was a visible abrupt jump of 156 W·m<sup>-2</sup> at the lower station within 15 minutes. For the night 9-10 September 2023, the fog also started to develop around midnight (at 00:50 the NET radiation was -5.8 W·m<sup>-2</sup>). NET radiation at SolarAOT<sub>lower</sub> became positive after sunrise, and a jump of 134 W·m<sup>-2</sup> occurred at the moment of fog disappearance at 6:20-6:40.

During the last night of observations, the NET radiation at the lower station was -44 W·m<sup>-2</sup> at midnight while at upper station -49 W·m<sup>-2</sup>. Starting at 02:00, the net radiation gradually increased from approximately  $-30 \cdot m^{-2}$  and reached  $0 \cdot m^{-2}$  at sunrise. The fog disappeared quickly, in less than ten minutes, at 05:35 the NET radiation changed by 125 W·m<sup>-2</sup>.

Having information on radiation fluxes at two levels allows for investigating the sensitivity of the change in LW radiation by the fog LWP. Fig. 15 shows the relationship between the modeled LWP content and the modeled difference in LW radiation

**Figure 15.** Relation between fog LWP (for days 9-11 Sep. 2023) and the difference of LW downwards flux between the upper and lower SolarAOT stations. The upper panel presents the radiation transfer model simulations and the lower panel corresponds to the balloon profiles (LWP) and radiometer observations at the upper and lower SolarAOT stations.

between the upper and lower stations. In addition to the modeled values, the lower panel of Fig. 15 shows the relationship for the data observed at both stations and the measured LWP on the balloon in the fog. In both graphs, the data exhibit a linear relationship, with Pearson correlation coefficients and corresponding p-values of -0.82 and 1.79e-10 for the model, and -0.37 and 0.02 for the observations, respectively. A straight line was fitted to the modeled data, given by the equation:

$$\Delta IR = -1.09 [W g^{-1}] \cdot LWP - 13.42 [W m^{-2}]$$
(17)

To the observed data was fitted a straight line:

$$\Delta IR = -0.74 [W g^{-1}] \cdot LWP + 12.25 [W m^{-2}]. \tag{18}$$

## 5 Conclusions







The purpose of this study was to capture vertical profiles of the microphysical and thermodynamic characteristics within fog layers, using in situ data collected by a tethered balloon during a field campaign. This article analyzes three cases of radiative fog that occurred in the valley of Strzyżów city (Southeastern Poland) in September 2023. In total, 74 soundings were performed, 41 of which included measurements with the OPC-N3 sensor, enabling the calculation of droplet size spectra. The three observed cases exhibited similar meteorological conditions, including temperature (T), relative humidity (RH), and aerosol scattering coefficient at 525nm (ASC $_{525}$ ). For each case, the liquid water path (LWP) exceeded 15 g·m $^{-2}$ , and in most instances, the fog did not transition into thick fog.

In the case of quasi-adiabatic boundary-layer clouds Brenguier et al. (2000), the droplet number concentration ( $N_c$ ) remains constant with height, and the increase in liquid water content (LWC) with height is primarily associated with the growth of droplet radii. Similarly, simulations of Atlantic stratocumulus conducted by Chang and Li (2002) showed that the increase in LWC with height was also linked to an increase in droplet size.

However, in the present study, which focuses on fog conditions, the observed variations in LWC were primarily associated with changes in  $N_c$ . This indicates that the increase in LWC is mainly due to the activation of fog droplets on new condensation nuclei, rather than droplet growth through the collision–coalescence process. Similar results were reported by Okita (1962); Egli et al. (2015), who also investigated the vertical distribution of microphysical properties in radiation fogs.

In the three presented cases of radiation fog over the Strzyżów valley, we observed that the effective radius (reff) decreases with height. This is consistent with Okita (1962) and partially with the observations of Egli et al. (2015) (in his study, reff decreased with height in some cases, but was mostly constant with height). As the fog developed, the decrease of  $r_{eff}$  with height became less pronounced, but remained visible.

In the work of Okita (1962), large droplets are mostly concentrated near the ground (the volume radius of large droplets decreases with height). In our study, drops larger than 18.5  $\mu$ m appear in the spectrum. DSDs with small concentrations of drops greater than 30  $\mu$ m were observed in experiments conducted by (Wendish et al., 1998; Gultepe et al., 2009; Mazoyer et al., 2022). These large droplets are the result of collision–coalescence and Ostwald ripening processes.

In the presented study, the volume of water is contained in two ranges of droplet radii: one around 4–5  $\mu$ m and the other above 18.5  $\mu$ m. Even though droplets with radii greater than 18.5  $\mu$ m are rare, the amount of water they carry is significant. The significance of the first range increases with height, while that of the second range decreases with height.

Simulations of numerical weather prediction (NWP) and large eddy simulations (LES) are predominantly based on bulk parametrization of e.g., LWP and  $N_c$  (Bergot et al., 2007; Khain et al., 2015). For improved NWP and LES simulations of fog formation and dissipation, it is essential to incorporate the droplet spectrum across the fog layer (Thoma et al., 2011). This would enable the removal of larger droplets through sedimentation, potentially alleviating the issue of excessively high LWC in fog forecasts (Philip et al., 2016; Pithani et al., 2019).

In the three analyzed cases, the fog dissipates from both the top and bottom. In the mature stage, the profiles of LWC and  $N_c$  increase with height and then, after reaching a maximum value, decrease toward the fog top height (CTH). During the

dissipation stage, the region above the maximum  $N_c/LWC$  becomes compressed. The maximum  $N_c/LWC$  is located above 80% of the fog height.

At the bottom of the fog, the smallest droplets evaporate. As no new droplets are formed, the larger droplets settle and are mostly located near the ground, which is reflected in higher  $r_{eff}$  values at the bottom of the fog. Droplets with radii up to 40  $\mu$ m can be described by an approximate formula for terminal velocity (u):

$$u(r) = k_1 \cdot r^2,\tag{19}$$

where  $k_1 \approx 1.19 \cdot 10^6 \text{ cm}^{-1} \cdot \text{s}^{-1}$  (Yau and Rogers, 1996). Using this formula, the fall velocity for drops larger than 18.5  $\mu$ m is 4.07 cm/s. In the absence of droplet growth and turbulence, for example, drops with a radius of 18.5 $\mu$ m will be removed within 7 minutes from a fog layer 100 m thick. During soundings conducted in the final stage of the fog life cycle, no large drops are observed, as they have already settled out.





In the article, we calculated the theoretical equivalent adiabaticity ( $\alpha_{eq}$ ). The values of  $\alpha_{eq}$  ranged between 0.0 and 0.6, similar to values previously reported for fog events. During one case, at the early stage of fog development, a negative value of  $\alpha_{eq}$  was observed (-3.2). This negative value was attributed to an elevated LWC near the ground. In the literature, cases where LWC decreases with height in fog are rarely observed Costabloz et al. (2024); Okita (1962); Boutle et al. (2018), and are typically associated with thin fog layers, with CTH not exceeding 40 m.

For the studied cases, the average fog-core LWC ranged from 0.2 to  $0.4 {\rm g \cdot m^{-3}}$ . LWC increased from the ground up to approximately 60% of CTH, and then decreased toward the fog top. The mean  $N_c$  reached up to 300 cm<sup>-3</sup>. In two fog cases (nights of 8–9 and 9–10 September), the mean  $N_c$  with height could be approximated by a parabolic curve. In the last fog case (10–11 September),  $N_c$  exhibited two local maxima at 25% and 88% of CTH. The mean reff in all cases was around 8–10  $\mu$ m and decreased linearly with height.

At the Strzyżów location, solar and infrared radiometers are installed at two different heights (within the fog layer and above it), enabling assessment of the fog's impact on radiation fluxes. A negative correlation has been shown between the difference in infrared radiation and the total water content in the fog (Pearson correlation coefficients and corresponding p-values of -0.82 and 1.79e-10 for the model, and -0.37 and 0.02 for the observations, respectively). The dissipation of fog can drastically change the total radiation flux at the ground within 10-30 minutes, by up to  $160 \text{ W} \cdot \text{m}^{-2}$ .

During fog, the mean bias between observed and modeled radiation fluxes is approximately  $2 \text{ W} \cdot \text{m}^{-2}$  for shortwave (SW) and  $11 \text{ W} \cdot \text{m}^{-2}$  for longwave (LW) radiation at the lower station. The good agreement of radiative fluxes indicates the consistency of the measurement data on the physical properties of fog.

Code and data availability. The data used in this article were uploaded to the repository. Nurowska, Katarzyna, 2024, "Microphysical and optical data of radiation fog in Strzyżów Valley, Poland", https://danebadawcze.uw.edu.pl/privateurl.xhtml?token=30df09f8-ce75-4c28-83ee-53bdc1b1c4fc, Dane Badawcze UW, V1.

Author contributions. K.N - Conceptualization, Methodology, Formal analysis, Investigation, Writing — original draft preparation, Writing—review and editing, Visualization; P.M. - lidar data visualisation; K.M. - Conceptualization, Supervision; All authors have read and agreed to the published version of the manuscript.

Competing interests. The authors declare no conflict of interest.

Acknowledgements. We thank the AERONET team, principal investigators and other participants for their effort in establishing and maintaining the network. This research is part of the OPUS project (Polish Grant No. 2017/27/B/ST10/00549) *Impact of aerosol on the microphysical, optical and radiation properties of fog*, which was funded by the National Science Centre, coordinated by the Institute of Geophysics, Faculty of Physics, University of Warsaw.

# List of abbreviations:


| Abbreviation         | Description                                       | Abbreviation     | Description                                   |
|----------------------|---------------------------------------------------|------------------|-----------------------------------------------|
| AE                   | aerosol Ångström exponent                         | MSE              | mean bias error                               |
| $lpha_{eq}$          | theoretical equivalent adiabaticity               | MWR              | microwave radiometer                          |
| AOD                  | aerosol optical depth                             | NTSB             | American National Transportation Safety Board |
| $\mathrm{ASC}_{525}$ | aerosol scattering coefficient of light at 525 nm | LES              | large eddy simulations                        |
| CCN                  | cloud condensation nuclei                         | LW               | longwave radiation                            |
| CTH                  | cloud / fog top height                            | LWC              | liquid water content                          |
| DSD                  | droplet size distribution                         | LWC <sub>0</sub> | non-zero surface liquid water content         |
| eBC                  | equivalent of black carbon                        | LWP              | liquid water path                             |
| $N_c$                | droplet number concentration                      | RH               | relative humidity                             |
| NC                   | particle number concentration                     | RMSE             | root mean square error                        |
| NWP                  | numerical weather prediction                      | SBL              | stable boundary layer                         |
| p                    | pressure                                          | SSA              | aerosol single scattering albedo              |
| PM                   | particle matter                                   | SW               | shortwave radiation                           |
| $r_{\rm eff}$        | effective radius                                  | T                | temperature                                   |
| RH                   | relative humidity                                 | vDSD             | volume droplet size distribution              |

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

**Figure A1.** Temporal variability of 510M lidar range corrected signal (at 532 nm) from the level of the upper station up to 12 km between 9 and 11 Sep. 2023.

Zhou, B., Du, J., Gultepe, I., and Dimego, G.: Forecast of Low Visibility and Fog from NCEP: Current Status and Efforts, Pure and Applied Geophysics, 169, 895–909, https://doi.org/10.1007/s00024-011-0327-x, 2012.

## Appendix A: Temperature set up in the model

The data about T and RH were taken from HYT and GY-63 and interpolated to the levels of the model. As the measurements were mostly reaching 115 m, above the temperature and humidity profile was set according to the measurement from atmospheric sounding from Tarnów (WMO station 12575) done by IMGW (Polish Institute of Meteorology and Water Management). Tarnów is 60 km in a straight line from Strzyżów city. For a better description of the merging of data sounding from Tarnów will be called sounding T.

To have a smooth transition between balloon sounding and the soundingT, the data from the balloon and soundingT were stitched together. For this purpose, first, the points were extrapolated so that at the stitching point, the values from both soundings were available. The procedure was performed for the last three points (highest points) from the balloon profile and the first two points of soundingT above the balloon profile. Then, for these five points, the average value weighted by the distance of the points was taken. As the soundingT does not always reach 10km, above soundingT the standard atmosphere profile for mid-latitude summer was used.

As the profile of RH was stitched and interpolated, the precipitable water (PW) was changed. To fix this issue, the whole profile of RH has been rescaled in such a way that PW is the same as the PW obtained from sounding T before interpolating.

## **Appendix B: Additional Tables and Figures**

| sounding nr | Day | Time          | Radiosonde | AE-51 | GY63        | HYT         | OPC-N3      | SPS30    | sounding nr | Day | Time          | Radiosonde | AE-51 | GY63        | HYT      | OPC-N3      | SPS30 | sounding nr | Day | Time          | Radiosonde | AE-51 | GY63     | HYT         | OPC-N3      | SPS30 |
|-------------|-----|---------------|------------|-------|-------------|-------------|-------------|----------|-------------|-----|---------------|------------|-------|-------------|----------|-------------|-------|-------------|-----|---------------|------------|-------|----------|-------------|-------------|-------|
| 1           | 08  | 20:29 - 20:42 |            |       |             |             |             |          | 28          | 09  | 17:59 - 18:10 | •          | •     |             |          |             |       | 54          | 11  | 02:11 - 02:17 |            |       | <b>•</b> | <b>•</b>    | <b>•</b>    | •     |
| 2           | 08  | 21:00 - 21:35 |            |       |             |             |             |          | 29          | 10  | 00:41 - 00:51 |            |       |             |          |             |       | 55          | 11  | 02:19 - 02:26 |            |       |          |             |             |       |
| 3           | 08  | 22:24 - 22:34 |            |       |             |             |             |          | 30          | 10  | 00:53 - 01:05 | •          |       |             |          |             |       | 56          | 11  | 02:29 - 02:37 |            |       |          |             |             |       |
| 4           | 09  | 00:22 - 00:28 |            |       | <b>&gt;</b> | <b>&gt;</b> | <b>&gt;</b> | •        | 31          | 10  | 01:18 - 01:28 |            |       |             |          |             |       | 57          | 11  | 02:43 - 03:02 | •          |       |          |             |             |       |
| 5           | 09  | 00:30 - 00:40 |            |       | <b>•</b>    | <b>•</b>    | •           | •        | 32          | 10  | 01:37 - 01:46 |            |       |             |          |             |       | 58          | 11  | 03:10 - 03:20 |            |       | •        | <b>•</b>    | <b>•</b>    | •     |
| 6           | 09  | 00:48 - 01:01 | •          |       |             |             |             |          | 33          | 10  | 01:55 - 02:04 | •          |       |             |          |             |       | 59          | 11  | 03:23 - 03:33 |            |       | <b>•</b> | <b>•</b>    | <b>•</b>    | •     |
| 7           | 09  | 01:20 - 01:27 |            |       | <b>•</b>    | <b>&gt;</b> | <b>•</b>    | •        | 34          | 10  | 02:24 - 02:34 |            |       | <b>•</b>    | <b>•</b> | <b>•</b>    | •     | 60          | 11  | 03:36 - 03:46 |            |       | <b>•</b> | <b>&gt;</b> | <b>•</b>    | •     |
| 8           | 09  | 01:32 - 01:41 |            |       | <b>•</b>    | <b>•</b>    | <b>•</b>    | •        | 35          | 10  | 02:40 - 02:48 |            |       | <b>•</b>    | <b>•</b> | <b>•</b>    | •     | 61          | 11  | 03:49 - 03:59 | •          |       |          |             |             |       |
| 9           | 09  | 02:34 - 02:42 |            |       | <b>•</b>    | <b>&gt;</b> | <b>•</b>    | •        | 36          | 10  | 02:54 - 03:06 |            |       | <b>•</b>    | <b>•</b> | <b>•</b>    | •     | 62          | 11  | 04:04 - 04:16 |            |       | <b>•</b> | <b>•</b>    | <b>•</b>    | •     |
| 10          | 09  | 02:45 - 02:55 |            |       | <b>&gt;</b> | <b>•</b>    | <b>&gt;</b> | •        | 37          | 10  | 03:09 - 03:23 | •          |       |             |          |             |       | 63          | 11  | 04:19 - 04:36 |            |       | <b>•</b> | <b>•</b>    | <b>&gt;</b> | •     |
| 11          | 09  | 03:03 - 03:27 | •          |       |             |             |             |          | 38          | 10  | 03:33 - 03:44 |            |       | <b>&gt;</b> | <b>•</b> | <b>•</b>    | •     | 64          | 11  | 04:40 - 04:56 |            |       | <b>•</b> | <b>•</b>    | <b>&gt;</b> | •     |
| 12          | 09  | 03:34 - 04:03 |            |       | <b>&gt;</b> | <b>&gt;</b> | <b>&gt;</b> | •        | 39          | 10  | 03:49 - 04:00 |            |       | <b>&gt;</b> | <b>•</b> | <b>•</b>    | •     | 65          | 11  | 04:59 - 05:15 | •          |       |          |             |             |       |
| 13          | 09  | 04:09 - 04:22 |            |       | <b>&gt;</b> | <b>•</b>    | <b>&gt;</b> | •        | 40          | 10  | 04:06 - 04:17 |            |       | <b>&gt;</b> | <b>•</b> | <b>•</b>    | •     | 66          | 11  | 05:19 - 05:30 |            |       | <b>•</b> | <b>•</b>    | <b>&gt;</b> | •     |
| 14          | 09  | 05:10 - 05:21 |            |       | <b>•</b>    | <b>&gt;</b> | <b>•</b>    | •        | 41          | 10  | 04:20 - 04:40 | •          |       |             |          |             |       | 67          | 11  | 05:32 - 05:45 |            |       | <b></b>  | <b>&gt;</b> | <b></b>     | •     |
| 15          | 09  | 05:23 - 05:34 |            |       | <b>&gt;</b> | <b>•</b>    | <b>&gt;</b> | •        | 42          | 10  | 04:50 - 05:00 |            |       | <b>&gt;</b> | <b>•</b> | <b>•</b>    | •     | 68          | 11  | 05:47 - 05:59 |            |       | <b>•</b> | <b>•</b>    | <b>&gt;</b> | •     |
| 16          | 09  | 05:37 - 05:45 |            |       | <b>•</b>    | <b>&gt;</b> | <b>•</b>    | •        | 43          | 10  | 05:06 - 05:21 |            |       | <b>•</b>    | <b>•</b> | <b>&gt;</b> | •     | 69          | 11  | 06:02 - 06:11 | •          | •     |          |             |             |       |
| 17          | 09  | 05:47 - 05:54 | <b>A</b>   |       | <b>A</b>    | <b>A</b>    | <b>A</b>    | •        | 44          | 10  | 05:25 - 05:37 |            |       | <b>&gt;</b> | <b>•</b> | <b>•</b>    | •     | 70          | 11  | 06:18 - 06:33 | •          | •     |          |             |             |       |
| 18          | 09  | 05:56 - 06:06 | <b>A</b>   |       | <b>A</b>    | <b>A</b>    | <b>A</b>    | •        | 45          | 10  | 05:40 - 05:56 | •          |       |             |          |             |       | 71          | 11  | 06:39 - 06:53 |            |       | <b>•</b> | <b>•</b>    | <b>&gt;</b> | •     |
| 19          | 09  | 06:09 - 06:17 | <b>A</b>   |       | <b>A</b>    | <b>A</b>    | <b>A</b>    | •        | 46          | 10  | 06:03 - 06:13 |            |       |             |          |             |       | 72          | 11  | 06:56 - 07:14 | •          | •     |          |             |             |       |
| 20          | 09  | 06:19 - 06:28 | <b>A</b>   |       | <b>A</b>    | <b>A</b>    | <b>A</b>    | •        | 47          | 10  | 06:23 - 06:31 |            |       | <b>&gt;</b> | <b>•</b> | <b>•</b>    | •     | 73          | 11  | 07:15 - 07:29 | •          | •     |          |             |             |       |
| 21          | 09  | 06:33 - 06:42 | <b>A</b>   |       | <b>A</b>    | <b>A</b>    | <b>A</b>    | •        | 48          | 10  | 06:31 - 06:40 |            |       | <b>&gt;</b> | <b></b>  | <b>•</b>    | •     | 74          | 11  | 07:36 - 07:51 | •          | •     |          |             |             |       |
| 22          | 09  | 06:45 - 06:53 | <b>A</b>   |       | •           | <b>A</b>    | •           | •        | 49          | 10  | 06:42 - 06:50 | •          | •     |             |          |             |       |             |     |               |            |       |          |             |             |       |
| 23          | 09  | 06:56 - 07:03 | <b>A</b>   |       | <b>A</b>    | <b>A</b>    | <b>A</b>    | •        | 50          | 10  | 06:52 - 07:00 |            |       |             |          |             |       |             |     |               |            |       |          |             |             |       |
| 24          | 09  | 07:07 - 07:13 | <b>A</b>   |       | <b>A</b>    | <b>A</b>    | <b>A</b>    | <b>A</b> | 51          | 10  | 07:01 - 07:19 | •          | •     |             |          |             |       |             |     |               |            |       |          |             |             |       |
| 25          | 09  | 07:18 - 07:41 | •          | •     |             |             |             |          | 52          | 10  | 07:11 - 07:17 |            |       |             |          |             |       |             |     |               |            |       |          |             |             |       |
| 26          | 09  | 07:42 - 07:56 | •          | •     |             |             |             |          | 53          | 10  | 07:22 - 07:32 | •          | •     |             |          |             |       |             |     |               |            |       |          |             |             |       |
| 27          | 09  | 07:57 - 08:08 | •          | •     |             |             |             |          |             |     |               |            |       |             |          |             |       |             |     |               |            |       |          |             |             |       |

**Table A1.** Apparatus used during each of the soundings of case study 9 - 11 Sep. 2023. Markers represent: ▶ - setup with OPC-N3, • - setup with radiosonde,  $\blacktriangle$  - setup with OPC-N3 and radiosonde,  $\Box$  - problems with collected data (sounding with partially recorded data were not taken into further analysis.

**Figure A2.** Visualization of radiation fog top. Photos were taken with the camera at the SolarAOT<sup>upper</sup> station at 4 UTC each day.

Figure A3. Figures present specific quantities measured by the balloon for each stage of fog observed during night 08-09 Sep. 2023.

**Figure A3.** Figures present specific quantities measured by the balloon for each stage of fog observed during night 08-09 Sep. 2023. From left: T from Vaisala radiosonde RS41, RH from Vaisala radiosonde RS41,  $N_c$ ,  $r_{eff}$  within the fog. Each colored line represents one balloon profile. The black thick line represents the mean of all the soundings, the colored area represents the range between +/- standard deviation from the mean. At the T plot dotted line presents the lapse rate, dashed red line presents the linear fit to T from 2 m to the height of maximum mean LWC. At the LWC plot dotted line presents the LWC adiabatic lapse rate, the dashed red line presents the linear fit to LWC from 2 m to the height of maximum mean LWC.

(b) Mature state of fog.

Figure A4. Figures present specific quantities measured by the balloon for each stage of fog observed during night 10-11 Sep. 2023.

**Figure A4.** Figures present specific quantities measured by the balloon for each stage of fog observed during night 09-10 Sep. 2023. From left: T from Vaisala radiosonde RS41, RH from Vaisala radiosonde RS41,  $N_c$ ,  $r_{eff}$  within the fog. Each colored line represents one balloon profile. The black thick line represents the mean of all the soundings, the colored area represents the range between +/- standard deviation from the mean. At the T plot dotted line presents the lapse rate, dashed red line presents the linear fit to T from 2 m to the height of maximum mean LWC. At the LWC plot dotted line presents the LWC adiabatic lapse rate, the dashed red line presents the linear fit to LWC from 2 m to the height of maximum mean LWC.

Figure A5. Figures present specific quantities measured by the balloon for each stage of fog observed during night 10-11 Sep. 2023.

Figure A5. Figures present specific quantities measured by the balloon for each stage of fog observed during night 10-11 Sep. 2023. From left: T from Vaisala radiosonde RS41, RH from Vaisala radiosonde RS41, N<sub>c</sub>, r<sub>eff</sub> within the fog. Each colored line represents one balloon profile. The black thick line represents the mean of all the soundings, the colored area represents the range between +/- standard deviation from the mean. At the T plot dotted line presents the lapse rate, dashed red line presents the linear fit to T from 2 m to the height of maximum mean LWC. At the LWC plot dotted line presents the LWC adiabatic lapse rate, the dashed red line presents the linear fit to LWC from 2 m to the height of maximum mean LWC.

Figure A6. Profiles with height of vDSD for 9-11 Sep. 2023 fog occurrence. In columns different days, in rows from top: beginning, mature, and disappearing stage of fog.