# Peer review of "Measurement report: Microphysical and optical characteristics of radiation fog - a study using in-situ, remote sensing, and balloon techniques."

_EGUsphere, 2024_

## Author Comment (AC1)

**Reply on RC1**

Dear Reviewer,
We would like to sincerely thank you for your thorough and constructive review of our manuscript. All of your comments and suggestions have been carefully considered and addressed in the revised version. We greatly appreciate the time and effort you invested in evaluating our work, as well as the valuable feedback that has helped us improve the quality of the manuscript.

Please find below our detailed responses to each of your comments.

**Major Comments**

1. **Question:** The radiation calculations. The greater part of the paper is concerned with presenting the observations themselves. It is useful to have more observations of (radiation) fog, especially when they include profiles of microphysical quantities. These, I think, largely stand on their own and make the paper of interest to a broad audience.

   A less significant part of the paper consists of a comparison of the observed radiative fluxes with fluxes calculated with a radiation code using the observed fog properties. As a check on closure, as noted on L18-19, this is useful, but I would question what is being checked here. The implication of this statement is that the microphysical measurements are being used to check the radiation ones. Especially given the large adjustments made during calibration in Eqs. 1-3, I would argue that the uncertainties in the microphysical measurements are greater than those in the radiative fluxes, so it is more of a check of the consistency of the microphysical measurements and their processing.

   **Answer:** We thank the referee for highlighting this point. As correctly noted, the main focus of the manuscript is the presentation of observational data collected during the campaign. To assess the consistency between the microphysical measurements and the observed radiative fluxes, we performed a radiative closure. The purpose of this closure was to validate whether the retrieved microphysical properties—together with the calibration of the measurement instruments—are consistent with the radiative fluxes observed by radiometers.

   Given that the OPC-N3 is a low-cost optical particle counter, we place greater confidence in the radiative flux measurements obtained from radiometers installed at the upper and lower stations. Since we did not have an independent method to verify the representativeness of the vertical microphysical profiles, we used these measurements as input for radiative transfer simulations. This approach allowed us to evaluate whether the OPC-N3 measurements are consistent with the observed shortwave fluxes. Nevertheless, we agree with the Reviewer that the uncertainties of

the radiation flux measurements are lower than those of the fog microphysical parameters.

To address the referee's concern that the purpose of the radiative closure may have been unclear, we have revised the manuscript to provide a more detailed and precise explanation.

2. **Question:** The separation of the discussion of the initialization of the radiation code in section 3.3 and the discussion of the results make it difficult for the reader to follow the argument. Some details are missing; for example, what was the surface albedo and how was the asymmetry parameter of the aerosol determined?

**Answer:**
We thank the referee for this helpful suggestion. The initialization of the radiation model was described in the methodology section (Section 3.3). However, we agree that some key details may not have been sufficiently emphasized or linked to the results discussion.

To improve clarity, we have ensured that all relevant model input parameters—including the surface albedo value and the method used to determine the aerosol asymmetry parameter—are explicitly stated in Section 3.3. We have also added clearer cross-references in the results section to guide the reader back to the model setup as needed.

In response to the specific points:

- The model allows for the specification of surface albedo based on the International Geosphere-Biosphere Programme (IGBP) land cover classification, using one of 20 predefined surface types. For all simulations performed in this study, the IGBP class was set to **"grassland"** (IGBP= 10), as the measurement site was located on a valley slope predominantly covered with grass, with sparse one-family houses.

  Surface albedo was implemented as a spectrally-resolved, solar-zenith-dependent parameter. The model computes 15-band shortwave spectral albedos based on lookup tables derived from MODIS albedo data and corrects them for the cosine of the solar zenith angle (u0) using an empirical scene-specific adjustment factor. The broadband albedo is then calculated as a weighted sum of the spectral albedos, with weights determined by the water vapor content (wv) and u0 according to the function, which approximates the spectral distribution of incoming solar radiation under clear-sky conditions.

  This approach enables accurate estimation of both spectral and broadband surface albedo for varying solar geometries and atmospheric moisture content. As the model was executed repeatedly for different times of day, both u0 and wv were dynamically updated for each run to reflect changes in solar position and atmospheric conditions, allowing for diurnal variation in surface radiative fluxes.

- The aerosol asymmetry parameter was determined for water droplets in the fog. It was derived using Mie scattering theory. Initially, the liquid water content and effective droplet radius were employed to estimate the droplet number concentration. Subsequently, spectral optical properties—extinction,

scattering, and single scattering albedo—were computed across relevant wavelengths. Finally, the asymmetry parameter was calculated by integrating the angular scattering phase function obtained from classical Mie theory.

We hope these changes will make the flow of the argument easier to follow.

3. **Remark:** In the case of LW radiation, the variation of the downward flux through the day is relatively small (approximately 330 - 370 Wm-2), but the temperature profile is obtained from soundings made at a different location at only two times during the day. This leads to uncertainties that are comparable in size to the variations in the fluxes and makes the regressions shown in Eqs. 21 and 22 rather meaningless. I would argue that the absolute values of the LW fluxes are poorly constrained, and that attention should be focused on the differences between the two local observing sites, indicating the impact of the fog layer.

   **Answer:** This is a good proposition for the improvement of the figure's informativeness. For the LW radiation, we changed the figure to represent the difference between the upper and lower stations for LW radiation in the fog.

4. **Remark:** Numerous small corrections need to be made in the text. Whilst many of them are trivial and can be corrected by careful copy-editing, there are a number that should be corrected at source, such as the sentence on L128 that is almost identical to that on L125, or the disagreement between Eq. 11 and the equivalent equation in Fig. 7. In a few cases, the meaning was unclear. I have tried to provide a fairly full list in the detailed comments below, but it is not exhaustive.

   **Answer:** We are grateful for your detailed comments at the grammatical level. All were carefully checked and corrected. We tried to give comprehensive explanations in places where the meaning could be unclear.

**Detailed Comments**

The comments were corrected in the text. Below are listed answers to some issues that needed more explanation:

1. **Remark:** Section 3.1 repeats material from Section 2.3

   **Answer:** This has been corrected so that Section 2.3 now describes only the instrumentation mounted on the balloon. Section 3.1 has been revised to focus on the flight methodology and the different configurations used during the balloon launches.

2. **Question:** Why are the differences so large? https://amt.copernicus.org/articles/16/2415/2023/ suggests a factor of 2. It would be useful to include a brief explanation of the origin of the calibration factor for readers who do not consult Nurowska et al. (2023).

**Answer:** OPC-N3 devices are considered low-cost sensors, which means that two identical units may not yield consistent results due to device-to-device variability. Therefore, cross-calibration between sensors or calibration against a reference-grade instrument is necessary to ensure measurement accuracy. Additionally, individual OPC-N3 units may exhibit signal drift over time, requiring periodic recalibration to maintain data reliability.

For this reason, it was not possible to directly use the calibration parameters provided in \citep{2023Nurowska}. Instead, the calibration had to be repeated following the methodology described in that work, to ensure compatibility with the specific sensors used in this study.

3. **Question:** I wondered how you chose the resolution for the radiation calculations. In practice, so long as you are interested in the fluxes at the top and bottom of the fog layer, rather than heating rates within the fog, this is probably not too crucial.

   **Answer:** The model is subject to constraints regarding the number of vertical levels that can be specified. We acknowledge that a vertical resolution of 10 m within the fog layer imposes limitations. However, since the primary focus of this study is on the radiative fluxes at the top and bottom boundaries of the fog, rather than on resolving the detailed vertical distribution of heating rates, we agree that this resolution is sufficient for our objectives, as you have noted.

4. **Question:** It is not clear how you have chosen the constant value. Whilst the thermal wavelengths are a bit too large for geometrical optics to apply, you could calculate a mean value of the effective radius by using the result from geometric optics, tau= 3 LWP/(2 rho_w r_e), so LWP/mean r_e should be equal to the integral of LWC/r_e through the cloud.

   **Answer:** The model employed in this study assumes that the droplet radius remains constant with altitude, which constitutes a known limitation. To determine the mean effective radius (r_eff) representative of the observed atmospheric profile, we calculated an average over altitude using measurements from two vertical soundings (ascent and descent). These data were obtained using the OPC-N3 optical particle counter. It should be noted that the model does not utilize the expression tau = 3 LWP/(2 rho_w r_e) as suggested. Instead, the optical properties of aerosols are computed explicitly using the Lorentz-Mie theory for spherical water droplets.

5. **Question:** The more general question is when you would regard a fog as thick

   **Answer:** As noted by Costabloz et al. (2024), different authors propose different thresholds for the transition from thin to thick fog, based on radiative, thermodynamic, geometric, and microphysical parameters. In their study, they examined five criteria to better assess the uncertainty associated with defining the transition time. These criteria are: (1) longwave net radiation approaching zero, (2) a negative temperature gradient between 50 m and 25 m, (3) turbulent kinetic energy (TKE) greater than $0.10\,\mathrm{m^2\,s^{-2}}$, (4) cloud top height (CTH) exceeding 110 m, and (5) liquid water path (LWP) greater than $15\,\mathrm{g\,m^{-2}}$.

In all thick fog cases analyzed by Costabloz et al., these conditions were met, but the time at which each criterion was fulfilled could vary by up to one hour. Therefore, it is challenging to determine the precise moment of transition; however, once the fog is fully developed into thick fog, all criteria are typically satisfied simultaneously.

In our observations, the condition of LWP > 15 g m$^{-2}$ was quickly met in all cases. For the event on 9 September, additional criteria were satisfied: first, the IR net radiation condition was met, followed by the CTH threshold. This indicates that a transition from thin to thick fog had begun. However, this process was interrupted by sunrise and did not fully develop into persistent thick fog.

For the fog events on the nights of 10 and 11 September, none of the other criteria were met later in time, and thus we conclude that a transition to thick fog did not occur during these cases.

Apart from the changes made in response to the comments from Referee 1 and Referee 2, we have also improved the preprocessing of the radiometer data. Specifically, short spikes in the signal—likely caused by transient obstructions such as birds—were removed using a filtering algorithm. Furthermore, the radiometric signal was smoothed using a 10-minute running mean.
It is also possible that water condensation occurred on the lower longwave (LW) radiometer during foggy conditions. We suspect such an event happened between 05:40 and 06:22 UTC on 11 September, during the dissipation of the fog. As a result, the corresponding flight conducted during this time has been excluded from the radiative closure analysis to ensure data quality.

---

## Author Comment (AC2)

**Reply on RC2**

Dear Reviewer,
 We are grateful for your careful reading of our manuscript and for the insightful comments and suggestions you provided. Your feedback has been extremely helpful in revising and refining the paper. We have taken all of your remarks into account and made corresponding changes in the revised version of the manuscript.

We respond to each of your comments in detail below.

**Major Comments**

1. **Remark:** The regression in Fig. 12h looks totally unconvincing, even though an equation is provided (equation 22). In fact, all equations in the text and in figures should be provided with both the value for Pearson's r and P. Note calculation of the P value takes into account the number of values used. Hence, a reasonable value for r (e.g. >0.5) is not necessarily associated with a relationship that can be distinguished from regressing random numbers (which can be inferred if e.g. $P < 0.05$).

   **Answer:** We agreed with the comment of the first reviewer, who noted that "the absolute values of the longwave fluxes are poorly constrained and that attention should be focused on the differences between the two local observing sites, indicating the impact of the fog layer." Therefore, Figure 12, which originally presented the longwave radiation for fog events, has been revised to show the differences between the two sites instead. For this updated version of the plot, the Pearson correlation coefficient is 0.83, and the corresponding p-value is 1.8×e-10. These values indicate a strong and statistically significant correlation.

2. **Remark:** Fog phases v fog stages: Line 191 refers to 'fog phases', but Lines 334 onwards describe 'fog stages'. Choose the terminology 'phase' or 'stage' and use this throughout.

   **Answer:** We have chosen the fog stages version, and it was consistently changed throughout the whole article.

3. **Remark:** Normally, visibility measured at 2 m above the ground is used to define the onset of fog, with fog defined as visibility <1 km. The authors subdivided fog events into phases/stages (Line 191), but they don't seem to have used their own visibility measurements. How do the fog development phases relate to visibility at 2 m (which would have been measured at the time of balloon launch by the TFMini instrument whenever the OPC-N3 was used – according to lines 181 and 184). I suggest a figure is added to the Appendices showing 2 m visibility data for the three fog events in relation to the fog stages - or add the visibility observations to Figure 5.

   **Answer:** The TFMini sensor was installed to assess its potential usefulness for visibility estimation. This sensor is typically employed in mechanical systems for

distance measurement. Deriving visibility information from this device requires further investigation, and at present, we do not plan to publish results, as its calibration for this purpose is challenging. Moreover, on the observation site, there was no dedicated instrument for direct visibility measurement. The division into fog stages was done based on the amount of LWP (exceeding or not exceeding $15\,\text{gm}^{-2}$).

A panel showing visibility has been added to Figure 5. The visibility estimate was derived using data from the ShadowGraph instrument. Specifically, the retrieved $r\_eff$ and LWC values were used to calculate visibility according to the Koschmieder formula.

4. **Remark:** The summaries of fog development for each night of observations (in Sections 4.2.2, 4.2.3 and 4.2.4) are very difficult to follow using the profiles shown in Figures 7, 8 and 9. This is because in the figures the same colour is used for every profile of the same variable. Different variables are shown with different colours. Instead it will be far easier for the readers to follow the written summaries of fog development if, for every variable there is a set colour for each stage of fog development (e.g. red for development, grey for mature, blue for disappearing).

   **Answer:** The colors in the Figures 7,8,9 has been changed. Each panel has the same colors palette, corresponding to different stages of fog evolution pink corresponds to the formation stage, blue to the mature stage, and yellow ochre to the dissipation stage.

5. **Remark:** Lines 566-567 'At the bottom of the fog, the smallest droplets evaporate.' What is the evidence for this? Evaporation of small droplets is feasible after sunrise from the top of fog layers, but not from the bottom. Instead, Weedon et al. (2024, QJRMS, https://doi.org/10.1002/qj.4702) argued that the inception of radiation fog is determined by the creation of suspended droplets, that is faster than their removal by occult deposition (direct deposition onto vegetation). Couldn't the small droplets at the bottom of the fog be removed progressively by occult deposition rather than evaporation?

   **Answer:** We acknowledge the reviewer's comment and appreciate the reference to Weedon et al. (2024), which provides valuable insights into the formation and evolution of radiation fog. In our observations, however, the situation appears somewhat different. As shown in the droplet spectra (Fig. A6) for the dissipation phase, droplets are present above 30 m, while no droplets are observed below this height. This phase occurred after sunrise, when solar radiation begins to supply energy to the surface, thereby inhibiting further droplet formation.

   While it is well established that in optically thick fog, droplet evaporation typically initiates from the top, our observations primarily concern optically thin fog layers. In such cases, solar radiation can more effectively reach the surface and lead to warming from below. We propose that, in these conditions, droplet evaporation may indeed occur at the base of the fog. Therefore, we suggest that in our observed case, evaporation near the fog base, rather than occult deposition, may explain the absence of small droplets at lower levels.

**Figure Comments**

We appreciate the reviewer's suggestions. The recommended changes to the figures have been made accordingly.

**Minor Comments**

We sincerely appreciate the reviewer's attention to detail. The minor comments have been addressed as recommended, and we are thankful for the valuable corrections that improved the grammar and overall language quality of the manuscript.

Apart from the changes made in response to the comments from Referee 1 and Referee 2, we have also improved the preprocessing of the radiometer data. Specifically, short spikes in the signal—likely caused by transient obstructions such as birds—were removed using a filtering algorithm. Furthermore, the radiometric signal was smoothed using a 10-minute running mean.

It is also possible that water condensation occurred on the lower longwave (LW) radiometer during foggy conditions. We suspect such an event happened between 05:40 and 06:22 UTC on 11 September, during the dissipation of the fog. As a result, the corresponding flight conducted during this time has been excluded from the radiative closure analysis to ensure data quality.

---

## Author Response (AR2)

**Response to the Editor:**

We would like to express our sincere thanks for your thoughtful feedback and helpful suggestions.

We appreciate your note regarding the length of the abstract. In response, we have revised and significantly shortened it in the new version of the manuscript to better align with journal expectations.

We also took the opportunity to make a minor revision to the Conclusions section, aiming to improve its structure and enhance readability for the reader, as per your valuable recommendation.

Thank you once again for your guidance and for considering our manuscript for publication.